# Integrated Approach for Synthetic Cathinone Drug Prioritization and Risk Assessment: In Silico Approach and Sub-Chronic Studies in *Daphnia magna* and *Tetrahymena thermophila*

**DOI:** 10.3390/molecules28072899

**Published:** 2023-03-23

**Authors:** Ariana Pérez-Pereira, Ana Rita Carvalho, João Soares Carrola, Maria Elizabeth Tiritan, Cláudia Ribeiro

**Affiliations:** 1TOXRUN—Toxicology Research Unit, University Institute of Health Sciences, IUCS-CESPU, CRL, 4585-116 Gandra, Portugal; 2Department of Biology and Environment, University of Trás-os-Montes and Alto Douro (UTAD), CITAB, 5000-801 Vila Real, Portugal; 3Inov4Agro—Institute for Innovation, Capacity Building and Sustainability of Agri-Food Production, 5000-801 Vila Real, Portugal; 4Interdisciplinary Center of Marine and Environmental Research (CIIMAR), University of Porto, Edifício do Terminal de Cruzeiros do Porto de Leixões, 4450-208 Matosinhos, Portugal; 5Laboratory of Organic and Pharmaceutical Chemistry, Department of Chemical Sciences, Faculty of Pharmacy, University of Porto, 4050-313 Porto, Portugal

**Keywords:** environmental management, psychoactive emergent contaminants, in silico prediction, protozoan, microcrustacean

## Abstract

Synthetic cathinones (SC) are drugs of abuse that have been reported in wastewaters and rivers raising concern about potential hazards to non-target organisms. In this work, 44 SC were selected for in silico studies, and a group of five emerging SC was prioritized for further in vivo ecotoxicity studies: buphedrone (BPD), 3,4-dimethylmethcathinone (3,4-DMMC), butylone (BTL), 3-methylmethcathinone (3-MMC), and 3,4-methylenedioxypyrovalerone (MDPV). In vivo short-term exposures were performed with the protozoan *Tetrahymena thermophila* (28 h growth inhibition assay) and the microcrustacean *Daphnia magna* by checking different indicators of toxicity across life stage (8 days sublethal assay at 10.00 µg L^−1^). The in silico approaches predicted a higher toxic potential of MDPV and lower toxicity of BTL to the model organisms (green algae, protozoan, daphnia, and fish), regarding the selected SC for the in vivo experiments. The in vivo assays showed protozoan growth inhibition with MDPV > BPD > 3,4-DMMC, whereas no effects were observed for BTL and stimulation of growth was observed for 3-MMC. For daphnia, the responses were dependent on the substance and life stage. Briefly, all five SC interfered with the morphophysiological parameters of juveniles and/or adults. Changes in swimming behavior were observed for BPD and 3,4-DMMC, and reproductive parameters were affected by MDPV. Oxidative stress and changes in enzymatic activities were noted except for 3-MMC. Overall, the in silico data agreed with the in vivo protozoan experiments except for 3-MMC, whereas daphnia in vivo experiments showed that at sublethal concentrations, all selected SC interfered with different endpoints. This study shows the importance to assess SC ecotoxicity as it can distress aquatic species and interfere with food web ecology and ecosystem balance.

## 1. Introduction

The aquatic environment is the destination for diverse classes of contaminants, including psychoactive recreational drugs [1,2]. Nevertheless, little is known about their impacts on wildlife and potential long-term effects at environmental concentrations on both non-target animals and humans. In the last two decades, new psychoactive substances (NPS) have emerged as a global threat to public health due to the increase in their consumption and the dynamics of the illicit market [3].

Recognizing which of these substances are truly a concern is a complex issue due to the lack of information about their pharmacology and potential toxicity because of the high number of new substances reported every year [4]. In 2020, the European Monitoring Centre for Drugs and Drug Addiction (EMCDDA) reported more than 820 NPS in the European drug market, and about 90 of them were detected for the first time between 2019 and 2020 [4]. The growing number of synthetic cathinones (SC) reflects the dynamic nature of the NPS market related to sales channels on the internet, social networks, and smartphone applications, leading to a fast and global phenomenon [5]. NPS are sold as an alternative to other recreational drugs such as cannabis, cocaine (COC), and 3,4-methylenedioxymethamphetamine (MDMA) [4,6,7]. The continued diversification and use across Europe remain a public health and legal challenge [8], with the expectation to increase the number of compounds and levels in water bodies.

Within the NPS, special attention has been given to SC such as mephedrone or 4-methylmethcathinone (4-MMC), methylone (bk-MAP), methcathinone or ephedrone (EPH), pentedrone (PTD), butylone (BTL), 3,4-dimethylmethcathinone (3,4-DMMC), buphedrone (BPD), 3,4-methylenedioxypyrovalerone (MDPV), and 3-methylmethcathinone (3-MMC) [6,7]. SC are *β*-keto analogs of commonly abused substances such as cathinone (CATH), isolated from the *Khat* plant (*Catha edulis*), which produce similar effects to their non-keto analogs’ amphetamine-type substances (indirect agonists of dopamine, serotonin, and noradrenaline receptors) [9]. Like other NPS, new SC emerge in the illicit market mainly in the internet market but also the dark web marketplace “Dream Market” [10], raising diverse concerns about toxicity and ecotoxicity.

Among the various sources of contamination, the discharge of treated wastewater is the major source of potential environmental contaminants. Indeed, after consumption or direct discharge, these substances reach the sewage systems and are carried through the sanitation networks to the wastewater treatment plants (WWTP). Although recent advances in wastewater treatment increased the removal efficiency of hazardous contaminants, some contaminants are still not eliminated or removal rates are low [11,12]. Regarding SC, some studies have already reported their presence in influent samples, but there is little information about their degradation rates, effluents, or environmental levels [13,14,15,16]. Toxicity effects of NPS, including some SC, have also been reported on aquatic organisms. For instance, exposure to pyrovalerone (MPP; ranging from 113.00 to 11,310.00 µg L^−1^) caused changes in the swimming behavior of zebrafish larvae (*Danio rerio*) [17] and changes in oxidative status and reproductive parameters were observed in microcrustacean *Daphnia magna* exposed to methamphetamine at 0.05 and 0.50 µg L^−1^ [18], and growth inhibition was observed and in protozoan *Tetrahymena thermophila* exposed to (*S*)-ketamine (>5000.00 µg L^−1^) [19]. Understanding the impact of NPS on ecosystems is crucial to provide essential data for establishing environmental safety levels, environmental policies, and mitigation actions. Regarding SC, as both physicochemical and biological properties change frequently due to specific structural modifications to circumvent legislation [20,21], the prediction of their potential ecotoxicity by in silico and in vivo evaluation are crucial tasks for ecological risk assessment. Today, in silico approaches are often used in combination with other toxicity tests to evaluate the environmental risk. Based on experimental data, structure–activity relationships, scientific knowledge, and specific software tools can be used to predict the potential toxicity and, in some situations, to quantitatively predict the toxic dose or potency. This avoids the realization of numerous in vivo assays, following European legislation, namely the Directive 2010/63/EU, firmly based on the principle of the three Rs: to replace, reduce, and refine the use of animals (vertebrates and cephalopods) used for scientific purposes. In this study, 44 SC were targeted for in silico approaches and a group of five emerging SC was prioritized for further in vivo ecotoxicity studies. Five SC were selected, namely 3,4-DMMC, BTL, 3-MMC, BPD, and MDPV based on consumption levels, EMCDDA reports [4,22], and recent reports in wastewaters [1]. An integrated approach based on in silico data and short-term exposure using the protozoan *T. thermophila* (28 h growth inhibition assay) and the microcrustacean *D. magna* checking different indicators of toxicity across life stage (8 days sublethal assay at 10.00 µg L^−1^) were accomplished.

*D. magna* and *T. thermophila* are widely distributed in freshwater systems displaying an important role in food chains [23,24,25]. Both organisms are used in ecotoxicological studies due to their short life cycle, high sensitivity to a variety of chemicals, and relatively easy maintenance and manipulation in the laboratory [23,26]. Both organisms are recommended by the Organization for Economic Cooperation and Development (OECD) to evaluate the toxicity of chemicals [27,28]. Two short-term exposure studies were performed to assess the sublethal effects of the five SC on protozoan and microcrustacean and compared with in silico data.

## 2. Results

### 2.1. In Silico Study

Based on the recent reports by EMCDDA [4,7,22], Bade et al. (2022) [1] and Almeida et al. (2022) [29], 44 emergent SC were selected for the in silico approaches (Figure 1). The results from the in silico studies obtained by the Estimation Program Interface (EPI) Suite^TM^ program are shown in Table 1, as well as the predicted toxicity for the *Tetrahymena pyriformis* protozoan using the Toxicity Estimation Software Tool (TEST^TM^) program. Chemical Abstracts Service (CAS) registry numbers and Simplified Molecular Input Line Entry System (SMILES) notations were required for both in silico programs (Appendix A).

Both computational programs are quantitative structure–activity relationships (QSAR) models for toxicological predictions used to determine the potential adverse effects of chemical entities for environmental risk assessment. The toxicity of contaminants and their environmental fate are related to their chemical structure and intrinsic physical/chemical properties (e.g., polar surface area (PSA), water solubility (WSol), and log K_OW_). These properties affect transport, permeability, bioavailability, bioconcentration, and bioaccumulation [30,31]. Information about data such as log K_OW_ and bioconcentration factor (BCF) are required by international regulations such as the OECD guidelines [32], the United States Environmental Protection Agency (USEPA) criteria [33], and the Registration, Evaluation, Authorisation and Restriction of Chemicals (REACH) criteria [34]. Indeed, log K_OW_ is a very important parameter for predicting the distribution of a substance in environmental compartments (water, soil, air, and biota). If log K_OW_ is lower than 3, the substance has no potential to bioconcentrate in living organisms.

In Appendix A, a summary of the in silico data obtained for the 44 SC is available. For each parameter, the 44 SC are organized in increasing order of selected parameters (Appendix A). The current study prioritized five SC for the in vivo ecotoxicity studies, namely 3,4-DMMC, BTL, 3-MMC, BPD, and MDPV, based on in silico approach data, consumption levels, EMCDDA reports [4,22], and recent reports in wastewaters [1].

WSol predicted values for the 44 SC varied between 2.57 mg L^−1^ (4-MeO-α-POP) and 51,470.00 mg L^−1^ (CATH). For the five SC prioritized, the order of solubility ranged from 70.24 to 5819.00 mg L^−1^ with the following order: MDPV < 3,4-DMMC < BTL < 3-MMC < BPD (Table 1).

Log K_OW_ values were also very different among SC, ranging from 1.38 for CATH to 5.47 for 4-MeO-α-POP. The order of SC with log K_OW_ values ≥ 3.00 was: 4-MeO-α-POP > NPP > 4-MeO-α-PHPP ≈ 5-PPDI > α-BHP > 4-BrPVP > 4-FPHP > MPP > α-PHP > α-PIHP > 4-MPBP > 4-MeO-α-PVP ≈ α-PVP > MDPV ≈ α-PPP. Regarding SC selected for in vivo experiments predicted log K_OW_ values were similar showing the following order: 3,4-DMMC > BTL > 3-MMC > BPD, ranging from 2.34 to 2.94, except for MDPV with a log K_OW_ value of 3.97 (Table 1).

PSA values obtained ranged between 29.1 and 47.6 Å^2^ and the order for selected in vivo experiments was: 3,4-DMMC, 3-MMC, and BPD < MDPV < BTL (Appendix A). In addition, prediction of the half maximal effective concentration (EC_50_), the median lethal concentration (LC_50_), and the chronic effect values (ChV) can be scrutinized through in silico computational models contributing to the reduction of animal experiments and allowing us to prioritize chemicals for toxicity tests [31,35,36].

Overall, 4-MeO-α-POP > NPP > 4-MeO-α-PHPP ≈ 5-PPDI showed the highest predicted acute and chronic toxicity for the fish, daphnia, and green algae, while EPH > bk-MAP > MTP > CATH exhibited the lower toxicity for the same organisms (Table 1). Regarding predicted acute toxicity for the five SC used for the in vivo experiments, for fish and daphnia the following order was found: MDPV > 3,4-DMMC > 3-MMC > BPD > BTL, whereas for green algae, the order was MDPV > 3,4-DMMC > BPD > 3-MMC > BTL. LC_50_ values ranged between 2.675 (MDPV) and 23.181 mg L^−1^ (BTL) for fish and 0.401 (MDPV) to 2.895 mg L^−1^ (BTL) for daphnia (Table 1). For green algae, the EC_50_ values ranged between 0.210 (MDPV) and 2.178 mg L^−1^ (BTL). Predicted chronic toxicity the order of toxicity was the following for the three organisms MDPV > 3,4-DMMC > 3-MMC > BPD > BTL and values ranged from 0.077 (MDPV) to 1.171 mg L^−1^ (BTL) for fish, from 0.041 (MDPV) to 0.249 mg L^−1^ (BTL) for daphnia, and from 0.082 (MDPV) to 0.748 mg L^−1^ (BTL) for green algae (Table 1). These results showed the higher toxicity of MDPV and the lower toxicity of BTL for the three organisms (green algae, daphnia, and fish).

Predicted BCF and BAF values for the 44 SC were very different according to the trophic level and the Arnot-Gobas method (no. 1 or 2). In general, considering the Arnot-Gobas method no. 1 and the three trophic levels of fish, the higher BCF and BAF predicted values were obtained for α-BHP (ranged from 793.200 to 1263.000 L Kg^−1^ wet wt^−1^), while the lower values were obtained for CATH (ranged between 2.257 and 2.947 L Kg^−1^ wet wt^−1^). On the other hand, in the Arnot-Gobas method no. 2 for the upper trophic level, BCF and BAF ranged from 15,020.000 to 568,100.000 L Kg^−1^ wet wt^−1^ for 4-MeO-α-POP (highest values) and between 3.469 and 3.510 L Kg^−1^ wet wt^−1^ for CATH (lowest values). For the in vivo selected five SC, predicted BCF values considering biotransformation rate ranged between 20.170 (BPD) and 83.320 L Kg^−1^ wet wt^−1^ (MDPV) for the upper trophic level, between 14.540 (BPD) and 104.500 L Kg^−1^ wet wt^−1^ (MDPV) for the mid trophic level, and between 13.040 (BPD) and 110.500 L Kg^−1^ wet wt^−1^ (MDPV) for the lower trophic level. Also, reflecting the biotransformation rate, predicted bioaccumulation factor (BAF) values ranged from 20.170 (BPD) to 83.320 L Kg^−1^ wet wt^−1^ (MDPV) for the upper trophic level, from 14.540 (BPD) to 104.600 L Kg^−1^ wet wt^−1^ (MDPV) for the mid trophic level, and from 13.050 (BPD) to 111.900 L Kg^−1^ wet wt^−1^ (MDPV) for the lower trophic level (Table 1).

Predicted BCF and BAF values assuming a biotransformation rate of zero ranged between 24.210 (BPD) and 973.500 L Kg^−1^ wet wt^−1^ (MDPV) and between 25.150 (BPD) and 2146.000 L Kg^−1^ wet wt^−1^ (MDPV), respectively (Table 1). For both Arnot-Gobas methods, BCF and BAF values order were the same for all trophic levels, namely MDPV > 3,4-DMMC > BTL > 3-MMC > BPD. Also, both BCF and BAF values were higher in the Arnot-Gobas method which assumes a biotransformation rate of zero. MDPV is the SC that presents higher BAF and BCF values. This result is expected since MDPV shows a log K_OW_ greater than 3, and consequently, displays higher lipophilicity. Therefore, is expected to bioaccumulate in fish tissues.

For the protozoan *T. pyriformis*, NPP showed the highest predicted toxicity (1.11 mg L^−1^) whereas CATH showed the lowest predicted toxicity (148.21 mg L^−1^). Toxicity estimation order for the selected SC used for the in vivo experiments is MDPV > 3,4-DMMC > 3-MMC > BPD > BTL, ranging from 6.50 to 51.80 mg L^−1^ (Table 1). Similar to the other organisms, for the protozoan, MDPV showed the highest toxicity and BTL the lowest toxicity.

### 2.2. In Vivo Studies

#### 2.2.1. Short-Term Exposure Assays with *T. thermophila*

Results from the growth inhibition assay with the protozoan are shown in Figure 2 and Table 2.

Controls showed an OD decreases greater than 60% and the reference test with K_2_Cr_2_O_7_ showed the reliability of the assay (Appendix A). Determination of EC_50_ or EC_20_ was not possible as no relation between response and concentration was observed. Consequently, it was not feasible to determine the dose–response curves for *T. thermophila*. However, a growth inhibition effect was observed for MDPV, BPD, and 3,4-DMMC. No changes in growth inhibition were observed in organisms exposed to BTL whereas an increase in growth was noted for organisms exposed to 3-MMC (Figure 2, Table 2).

#### 2.2.2. Short-Term and Sublethal Exposure Assay with *D. magna*

No significant differences were found in mortality (<10%) for any of the SC. In fact, the results obtained for the daphnia mortality agree with the in silico data since all SC showed LC_50_ values greater than 401.00 µg L^−1^ (MDPV) for daphnia, and the concentration studied in our in vivo study was considerably lower (10.00 µg L^−1^). Effects of SC on the morphophysiological parameters of *D. magna* are shown in Figure 3 and summarized in Table 3.

BTL caused a significant increase in body size in both juveniles and adults (days 3 and 8, respectively). Different responses were found for juveniles and adults exposed to 3-MMC. Indeed, a significant increase in body size was observed for the juveniles at day 3, whereas a significant reduction of body size was observed in adults at day 8 (Figure 3, Table 3). A significant decrease in body size was also noted for MDPV in adults. No changes in body size were observed on both days for BPD and 3,4-DMMC (Figure 3, Table 3). SC also showed to interfere with heart area and size. Indeed, at day 3, a significant increase in heart area (except for BPD) and size of juveniles were observed for all SC. However, on day 8, only BTL, BPD, and 3,4-DMMC continued to stimulate heart area and size growth, whereas MDPV and 3-MMC caused a significant decrease in heart area and size (Figure 3, Table 3). Regarding heart rate, at days 3 and 8, a significant increase was observed for all SC except at day 3 in the organisms exposed to 3-MMC (Figure 3, Table 3).

No significant differences were observed in swimming speed and active time for all SC. However, total distance traveled was significantly increased in the organisms exposed to BPD and 3,4-DMMC (Figure 4, Table 4).

Regarding reproductive parameters, although a tendency to the increase in the number of eggs per daphnia was observed for all SC (except for 3-MMC), only exposure to MDPV showed a significant increase (Figure 5, Table 5).

A significant increase in reactive oxygen species (ROS) levels was verified for BTL and 3,4-DMMC exposures, but no changes were found for other 3-MMC, MDPV, and BPD (Figure 6, Table 6). A significant increase in thiobarbituric-acid-reactive substances (TBARS) levels was observed for MDPV and 3,4-DMMC (Figure 6, Table 6). No changes in catalase (CAT) and acetylcholinesterase (AChE) enzymatic activity were observed except for BPD, which caused a significant stimulation of AChE activity.

## 3. Discussion

Some studies reported the occurrence of SC in wastewaters as influents and effluents, but information about surface waters is scarce [13,14,37,38]. For instance, 4-MMC (up to 106.00 ng L^−1^), bk-MAP (up to 12.00 ng L^−1^), and MDPV (up to 6.00 ng L^−1^) were measured in influent wastewater samples from eight European countries [15]. Another study reported the presence of BTL, MDPV, BPD, and 3,4-DMMC in a low range (1.00 to 20.00 ng L^−1^) in urban wastewaters from WWTP from different European cities [12]. The occurrence of BPD and 3,4-DMMC in Portuguese surface waters and effluent samples was reported though below their limit of quantification of 125.00 and 250.00 ng L^−1^, respectively [16]. Therefore, in addition to the significant human health risks, assessment of their toxic effects on aquatic organisms has become important to provide scientific evidence about their ecotoxicity. In this study, 44 SC were selected for in silico studies, and a group of five emerging SC was prioritized for further in vivo ecotoxicity studies.

*In silico* tests were applied to predict the physicochemical properties (WSol and log K_OW_), the BCF, BAF and potential toxicity for diverse organisms from different trophic levels. The 3,4-DMMC, BTL, 3-MMC, BPD, and MDPV were selected for the further in vivo experiments based on in silico approaches data, consumption levels, EMCDDA reports [4,22], and recent reports in wastewaters [1].

The log K_OW_ is important to identify potential contaminants of concern according to OECD guidelines and USEPA criteria [32,33]. Predicted log K_OW_ values were very different among SC ranging from 1.38 for CATH to 5.47 for 4-MeO-α-POP. The order of SC with log K_OW_ values ≥ 3.00 was 4-MeO-α-POP > NPP > 4-MeO-α-PHPP ≈ 5-PPDI > α-BHP > 4-BrPVP > 4-FPHP > MPP > α-PHP > α-PIHP > 4-MPBP > 4-MeO-α-PVP ≈ α-PVP > MDPV ≈ α-PPP, showing the higher potential for bioconcentration and bioaccumulation. Predicted log K_OW_ values for SC selected for the in vivo assays were similar for all SC (< 3.00) except for MDPV, which showed the highest value (3.97) indicating a higher potential for bioaccumulation. The WSol values for the 44 SC varied between 2.57 for 4-MeO-α-POP, which showed the lower WSol, and 51,470.00 mg L^−1^ for CATH, with the higher WSol. Considering the five SC prioritized, values ranged from 70.24 to 5819.00 mg L^−1^ in the following order MDPV < 3,4-DMMC < BTL < 3-MMC < BPD, i.e., MDPV showing the lower WSol.

The 44 SC order of toxicity was similar for the three model organisms (fish, daphnid, and green algae). In general, the higher acute and chronic toxicity were obtained for 4-MeO-α-POP > NPP > 4-MeO-α-PHPP ≈ 5-PPDI, whereas the lower toxicity values were found for EPH > bk-MAP > MTP > CATH for the three organisms (green algae, daphnia, and fish). For the SC selected for the in vivo assays, the acute and chronic toxicity showed the highest toxicity of MDPV towards fish, daphnia, and green algae, whereas BTL showed the lowest toxicity. The susceptibility of organisms was different with green algae showing higher sensitivity to SC in comparison to fish and daphnia. Regarding chronic toxicity, daphnia showed higher vulnerability. Overall, *T. pyriformis* ICG_50_ predicted values showed higher toxicity for NPP (1.11 mg L^−1^) and the lower toxicity for CATH (148.21 mg L^−1^), whereas for the prioritized five SC, the predicted toxicity values ranged between 6.50 and 51.80 mg L^−1^ in the following order: MDPV > 3,4-DMMC > 3-MMC > BPD > BTL.

BCF and BAF values for the 44 SC varied according to the trophic level and the *Arnot-Gobas* method. Considering the Arnot-Gobas method no. 1 (three trophic levels of fish), the higher BCF and BAF predicted values were obtained for α-BHP, whereas the lowest were predicted for CATH. Regarding the Arnot-Gobas method no. 2 for the upper trophic level, the highest BCF and BAF predicted values were obtained for 4-MeO-α-POP, while the lower values were found for CATH. In the five SC selected for the in vivo ecotoxicity tests, and the predicted BCF and BAF values estimated the same order of potential for bioaccumulation for three trophic levels (MDPV > 3,4-DMMC > BTL > 3-MMC > BPD). According to the REACH criteria, substances can be categorized as very bioaccumulative (>5000 L Kg^−1^ wet wt^−1^), bioaccumulative (5000 ≥ 2000 L Kg^−1^ wet wt^−1^), or not bioaccumulative (<2000 L Kg^−1^ wet wt^−1^) [34]. Considering these criteria, SC are not considered bioaccumulative except MDPV (2146 L Kg^−1^ wet wt^−1^), which is categorized as a bioaccumulative substance. These results are in concordance with predicted log K_OW_ values and the chemical structures.

For a comprehensive study regarding the prioritization of the five SC’ short-term exposure, tests were performed with protozoan and daphnia and compared with in silico SC toxicity prediction.

Protozoan growth inhibition assay was accomplished from 1.25 to 40.00 mg L^−1^ to estimate EC_50_ values. Within this range of concentrations, no EC_50_ values were possible to be determined. Higher concentrations were not tested as they are not expected to be measured in environmental aquatic ecosystems. Nevertheless, protozoan growth response depended on the SC showing different susceptibilities to the substances. The highest growth inhibition was observed for MDPV followed by BPD and 3,4-DMMC. No toxicity was found for BTL and 3-MMC caused an increase in protozoan growth. The protozoan is not included in the ECOSAR^TM^ model organisms; however, toxicity was possible to obtain using TEST^TM^ program. Data showed higher toxicity of MDPV and lower toxicity of BTL. These models have been pointing out that toxicity increases with the increase in the number of atoms and degree of methylation per compound and that toxicity decreases with an increase in nitrogen substitution [39]. In vivo experiments with protozoan are in accordance with the in silico data that showed higher toxicity of MDPV and lower toxicity of BTL. However, 3-MMC showed to stimulate protozoan growth, whereas in silico predictive data indicated toxicity of this substance after 48 h of exposure. Similar results, i.e., growth increase instead of growth inhibition, have been reported for other organisms as bacteria exposed to environmental contaminants. These organisms may use the contaminants as sources of carbon that causes stimulation of growth. Mennillo et al. (2018) reported a growth increase at the highest concentration of ketoprofen on the bacteria *Vibrio fischeri* [40]. Differences between the in silico data and in vivo experiments can also be related to the time of exposure (28 h in the in vivo studies whereas 48 h for the in silico study).

Regarding daphnia, in silico data showed toxicity at high concentrations (mg L^−1^), with MDPV showing the higher toxicity and BTL the lower toxicity. As these concentrations are not expected to occur in the environment, a sublethal concentration, 10.00 μg L^−1^, was selected and different parameters as checkpoints of toxicity were evaluated. Indeed, in silico tests are based on acute immobilization and daphnia reproductive assays that can be insufficient to evaluate toxicity as other endpoints can be affected at lower concentrations and affect the survival of the organisms. Thus, morphophysiological, behavioral, reproductive, and biochemical parameters were evaluated as biomarkers of toxicity [41,42,43].

Morphophysiological parameters showed to be affected in a substance and daphnia life cycle dependent manner. BTL affected all morphophysiological parameters causing a significant increase in body size, heart area, heart size, and heart rate in both juveniles and adults. All other SC also stimulated heart rates in both juveniles (except 3-MMC) and adults. Regarding heart area and size, different responses were observed among SC. All SC affected heart size and area in juveniles, causing an increase except BPD in juveniles, but at day 8 a decrease in heart area and size was observed for both MDPV and 3-MMC. Studies of the effects of SC on the morphophysiological parameters of this microcrustacean are non-existent so far; however, interferences in morphophysiological parameters such as heart rate, thoracic limb activity, and mandible movements have been reported for other chemicals including psychoactive substances [43,44].

Behavioral parameters, such as swimming speed, distance traveled and active time, gained special attention since SC are NPS acting at the level of the central nervous system that directly affects the locomotive abilities of daphnia [41,42,43,45]. Although a tendency to the increase in swimming speed was observed for some SC (BTL, BPD, and 3,4-DMMC) no significant effects were noted. Additionally, no changes in active time were observed for any of the SC. Regarding total distance traveled, an increase was observed for BPD and 3,4-DMMC. Changes in swimming activity (distance moved and swimming speed) were observed in *D. magna* after 21 days of exposure to COC at 0.05 and 0.50 µg L^−1^ [45].

Only MDPV showed to interfere with first reproductive events causing a significant increase in the number of eggs per daphnid. Changes in reproduction events have been reported for other psychoactive drugs (namely, methamphetamine and COC) after 21 days of exposure [18,45], but the 21-day reproductive assay was not performed in our current work.

SC also exhibited distinct effects on biochemical parameters. Although BTL and 3,4-DMMC caused an increase in ROS levels, no changes were observed in CAT activity. Additionally, no changes were noted in TBARS levels for BTL, but 3,4-DMMC caused an increase in its levels corroborating the increase in ROS levels. Parolini et al. (2018) reported that benzoylecgonine induces oxidative stress on *D. magna* after 48 h of exposure at environmental concentrations (0.50 and 1.00 µg L^−1^) [46]. A tendency to increase in TBARS levels was also observed for all SC; however, significant changes were noticed for 3,4-DMMC and for MDPV. CAT is a relevant antioxidant enzyme, which acts in the protection of cells from ROS species, transforming hydrogen peroxide in oxygen and water. Although an increase was observed for ROS and TBARS, no changes in CAT activity were noted. Changes in CAT activity have been reported in daphnia exposed to psychoactive substances [18,47]. AChE is an enzyme that plays an important role in the normal regulation of the central nervous system, being an important biomarker [46]. Only organisms exposed to BPD showed significant alterations (increase) in AChE levels.

*In vivo* experiments with daphnia, at sublethal concentrations, showed that all SC can interfere with different endpoints, and therefore, it was not possible to observe a distinct SC toxicity. BTL interfered with all morphophysiological parameters in contrast to the lower in silico data toxicity potential, and thus toxicity can be underestimated. Therefore, care should be taken when using in silico data as contaminants can interfere with endpoints not considered in these programs.

## 4. Materials and Methods

### 4.1. Chemicals and Reagents

Purity of all SC standards was >98.5% and in the form of racemates (50.0% of each enantiomer). The 3-MMC and 3,4-DMMC were acquired from LGC Standards GmbH (Wesel, Germany); BPD was obtained from Cayman Chemical (Ann Arbor, MI, United States of America (USA)); BTL was purchased from Cerilliant (Round Rock, TX, USA), and MDPV was acquired from Lipomed AG (Arlesheim, Switzerland). Individual stock solutions of each SC for the in vivo assays were prepared at 1.00 mg mL^−1^ in 10 mL of ultrapure water (UPW; Ultrapure Water System (SG Ultra Clear UV plus)) and stored in amber bottles at −20 °C. Potassium dichromate (K_2_Cr_2_O_7_; ~98.0%) was obtained from José Manuel Gomes dos Santos, LDA (Portugal). For biochemical assays, bovine serum albumin (BSA; ≥96.0%) at 0.10 mg mL^−1^ in UPW, CAT from *Aspergillus niger* (≥4.00 units mg^−1^ protein) at 69,629.00 U mL^−1^, 2′,7′-dichlorofluorescein (DCF; ~90.0%) at 10.00 mM in dimethyl sulfoxide (DMSO; ≥99.9%) and malondialdehyde (MDA; ≥96.0%) at 5.00 mM in UPW were obtained from Sigma-Aldrich (St. Louis, MO, USA or Steinheim, Germany).

### 4.2. In Silico Study

For in silico studies, the EPI Suite^TM^ program (version 4.11, November 2012) [48] developed by the USEPA with KOWWIN^TM^, WSKOWWIN^TM^, ECOSAR^TM^, and BCFBAF^TM^ programs were used. The EPI Suite^TM^ program uses a single input to run diverse validated estimation programs allowing to predict log K_OW_, WSol, bioaccumulation, and estimate toxicity for fish (96 h and 14 days) [31,49,50], daphnia (48 h and 21 days) [31], and green algae (48 h) [51], and ChV for fish, daphnia, and green algae (*Chlorophyta*). ChV is defined as the geometric mean by the following equation:(1)$$\mathrm{ChV}=10^{\left[\frac{\log\left(LOEC\times NOEC\right)}{2}\right]}$$
where the *NOEC* is the no-observed-effect concentration and the *LOEC* is the lowest-observed-effect concentration.

For estimation of these parameters, the chemical names, CAS registry numbers, and SMILES notations were introduced on in silico computational programs. These parameters as well as PSA values and IUPAC (International Union of Pure and Applied Chemistry) names were obtained from PubChem website searches (https://pubchem.ncbi.nlm.nih.gov, accessed on 21 December 2022 and on 8 March 2023). For more detailed information, please see the Appendix A). Appendix A provides the acronyms, IUPAC names, CAS numbers, chemical structures, SMILES notations, and PSA values of the 44 SC selected for in silico studies.

The KOWWIN^TM^ program (version 1.68, September 2010) was used to estimate the log K_OW_ (log octanol-water partition coefficient) of chemicals using an atom/fragment contribution method [33,52]. The WSKOWWIN^TM^ program (version 1.42, September 2010) estimates the WSol of an organic compound using the log K_OW_ previously estimated by KOWWIN™ program and then applicable correction factors if needed [53,54,55]. The ECOSAR^TM^ (version 1.11, July 2012) estimates the aquatic toxicity of chemicals, namely acute toxicity and chronic toxicity to aquatic organisms such as fish, aquatic invertebrates, and green algae. BCFBAF™ program (version 3.01, September 2012) provides screening levels of BCF and BAF prediction by regression model based on log K_OW_ values and includes correction factors for biotransformation and ionization. The Arnot-Gobas method [56] was considered (including biotransformation rate estimates and assuming a biotransformation rate of zero) to calculate BCF and BAF for three trophic positions of fish. As no information for protozoan is possible to obtain using EPI Suite^TM^ program, the TEST^TM^ program (version 5.1.2, October 2022) was used [57]. This application was developed by USEPA to estimate toxicity values for several endpoints including the 48 h assay on the protozoan *T. pyriformis* by accessing the 50% of the growth inhibition concentration (IGC_50_).

### 4.3. Ecotoxicity Assays

#### 4.3.1. Sub-Chronic Assay with *T. thermophila*

Protozoan growth inhibition toxicity assay was performed based on the procedures described in the Standard Operating Procedures for Toxkit tests (Protoxkit F^TM^, MicroBioTests Inc., Gent, Belgium) and in accordance with OECD Guideline no. 244 [28]. The SC solutions and reference test with K_2_Cr_2_O_7_ were prepared in standard freshwater medium (SFM; 96 mg of NaHCO_3_, 120 mg of CaSO_4_2H_2_O, 123 mg of MgSO_4_7H_2_O and 4 mg of KCl in 1 L of distilled water). Individual SC stock solutions were prepared at 1.00 mg mL^−1^ in UPW and exposure solutions by dilution of the stock solution with SFM. The concentrations used were 1.25, 2.50, 5.00, 10.00, 20.00, and 40.00 mg L^−1^ for each SC. For the reference test, a stock solution of K_2_Cr_2_O_7_ at 100.00 mg L^−1^ in SFM was prepared and exposure concentrations by dilution with SFM. The reference test was performed at 5.60, 10.00, 18.00, 32.00, and 56.00 mg L^−1^. More details can be found in Appendix A (Appendix A). Each concentration was performed in triplicate.

#### 4.3.2. Sublethal Assay with *D. magna*

##### *D. magna* Culture Maintenance

Monoclonal cultures of *D. magna* were maintained under laboratory controlled conditions of light intensity (6000 lux), photoperiod (16:8 h light: dark), temperature (20 ± 2 °C) and kept in moderately hard reconstituted water (MHRW) [27]. Daphnia were maintained in groups of 30 individuals per 800 mL of MHRW, supplemented with a vitamin mixture (biotin, thiamine, and cyanocobalamin), algae extract (*Ascophyllum nodosum*, Extract Sol-Plex Sierra acquired from Alltech Naturally (Sintra, Portugal)) and *Saccharomyces cerevisiae* yeast acquired from Pura Vida (Lisbon, Portugal). Organisms were fed with a microalgae suspension of *Raphidocelis subcapitata* at 3.0 × 10^5^ cells mL^−1^ day^−1^ (neonates/juveniles) or 6.0 × 10^5^ cells mL^−1^ day^−1^ (adults). The culture media was renewed every 2 days. Microalgae were cultured in Woods Hole MBL medium, in a semicontinuous 4 L batch culture at a 16:8 h light: dark cycle (20 ± 2 °C) [58]. Culture maintenance details can be found in Appendix A. Individuals born between the 3rd and 5th brood with less than 24 h were used to start new cultures and to perform the experimental assays.

##### Experimental Design

Five replicates were used for the control and for each compound experiment. Each experimental unit contained 15 neonates per 250 mL of culture medium. From the individual SC stock solutions prepared at 1.00 mg mL^−1^ in UPW, individual intermediate solutions were prepared at 1.00 mg L^−1^ (in 100 mL of UPW) and stored at 4 °C, and used to prepare the concentration of 10.00 µg L^−1^ with MHRW for the exposure experiments. Every 2 days, the culture medium was renewed, and organisms were fed with a microalgae suspension.

Mortality was monitored over the exposure time. On days 3 and 8, the morphophysiological parameters were determined using a Zeiss Axiostar plus optical microscope (Carl Zeiss, Jena, Germany) coupled to a PowerShot G9 digital video camera (Canon, Tokyo, Japan). Three organisms were randomly collected from each replicate, photographed and video recorded for 1 min (min), and later analyzed to determine the body size, heart size and area, and heart rate. To assess morphometric parameters free image measurement software was used and DaVinci Resolve software (version 17.2 Build 11) was used to change clip speed (to 25% frame reduction) for heart rate determination. From day 8, photographs were taken and used for the determination of the number of eggs per daphnia.

On day 5, five individuals were randomly collected from each experimental replicate and placed into a 6-well plate with ~5 mL of the respective exposure medium, and video was recorded for 1 min with a digital video camcorder (Canon Legria HF R506, Japan). Each well was previously filled with 5 mL of melted 1% agarose and once solidified a circular swimming area of 27 mm was created using a plastic cylinder. These holes provide a swimming area with excellent optics and visibility for video recording. The clips were analyzed with TheRealFishTracker program to obtain swimming speed, active time, and total distance travelled [59,60].

After 8 days of exposure, the survival individuals were collected into an Eppendorf tube, washed twice with phosphate buffer solution (PBS; 0.800 g of NaCl; 0.020 g of KCl; 0.144 g of Na_2_HPO_4_ and 0.024 g of KH_2_PO_4_ in 100 mL of UPW, pH adjusted to 7.4), and then stored on 250 µL of PBS at −80 °C. Samples were homogenized using an ultrasonicate (Vibra-Cell^TM^ model VCX750 with a tip diameter of 3 mm, both from Sonics & Materials, Inc.), and then centrifuged at 15,000× *g* for 10 min at 4 °C (Heraeus Biofuge 1.0R refrigerated centrifuge (Hanau, Germany) for protein quantification and determination of AChE, CAT, TBARS, and ROS [44,45,46,61,62,63,64,65,66,67,68,69]. Oxidative stress and enzymatic activity were determined spectrophotometrically (each sample in duplicate) using a microplate BioTek Synergy plate reader 2 (Vermont, USA). More details are present in Appendix A.

### 4.4. Statistical Analysis

Protozoan growth inhibition assay data were analyzed in GraphPad (Version 8.0.1.244) using nonlinear regression with variable slope (four parameters) and least-squares fit method. To assess which concentrations caused significant growth inhibition, one-sample unidirectional *t*-tests (% growth inhibition > 0) were performed for each concentration and all substances (a total of 35 one-sample *t*-tests were performed); to overcome the multiplicity problem (testing related multiple hypotheses; [70]), *p*-values were adjusted by controlling the false discovery rate [71] using the *p*.adjust() function in R [72].

Data from sublethal assay on daphnia were obtained using Jamovi program (version 2.2.5), a free statistical software application [73]. To evaluate the five SC data effects, both general linear models (one-way ANOVA) and generalized linear models were applied [74,75]. General linear model fit by OLS was employed for survival, morphophysiological, behavioral, and biochemical parameters, while negative binomial generalized linear models (model for count data) were applied for reproductive events (number of eggs per daphnia). In both cases, the occurrence of a significant effect of SC was additionally examined with Dunnett contrasts to evaluate significant differences between treatment and control. The differences were considered statistically significant if the *p* < 0.05.

## 5. Conclusions

The presence of SC at a low ng L^−1^ range has been reported in WWTP effluents. However, the information available concerning the ecotoxicological effects on aquatic model species is still poorly explored.

For that, 44 SC were selected for the in silico studies and a group of five SC prioritized based on consumption and recent reports in wastewaters for an integrative in vivo study using two ecologically relevant organisms belonging to different trophic levels (protozoan and microcrustacean), namely BTL, BPD, MDPV, 3-MMC, and 3,4-DMMC. The in silico data revealed that MDPV is the SC with the most potential for fish bioaccumulation and also the most toxic (at acute and chronic levels) for the three organisms (fish, daphnia, and green algae). Additionally, different organism susceptibilities were obtained regarding both acute (green algae) and chronic (daphnia) evaluation. For *T. pyriformis*, MDPV showed the highest toxicity, while BTL showed the lowest toxicity.

Similar results were found for the in vivo experiments using the *T. thermophila*, namely increasing growth inhibition (MDPV > BPD > 3,4-DMMC), whereas no effects were observed for BTL. However, a stimulatory effect was observed for 3-MMC. Considering the sublethal ecotoxicity assays in the microcrustacean, different susceptibilities were noted depending on the type of SC and endpoints study.

Our work shows that SC do not affect mortality at the concentrations studied; however, they interfered at sublethal levels with several endpoints in *D. magna* (precisely, morphophysiology (BTL significantly increases the body size, heart size, heart area and heart rate in juveniles and adults daphnia), swimming behavior (BPD and 3,4-DMMC significantly raise the total distance traveled), reproduction (MDPV significantly increase the number of eggs per daphnia), and oxidative stress and biochemical activity (3,4-DMMC and MDPV significantly increase the TBARS levels)).

These results emphasize that, although in silico programs provide important data about the toxicity potential of diverse substances, in vivo experiments at sublethal or environmental reported levels focusing on other endpoints demonstrate that contaminant toxicity can be underestimated.

In this sense, further in vivo experimental studies should be conducted to extend the current knowledge about SC ecotoxicity effects on non-target aquatic organisms and to expand to other organisms belonging to different trophic levels, such as fish, improving ecotoxicological testing in risk assessment procedures. Additionally, the potential of mixtures should be considered as additive or synergistic effects can occur.

## Figures and Tables

**Figure 1 molecules-28-02899-f001:**
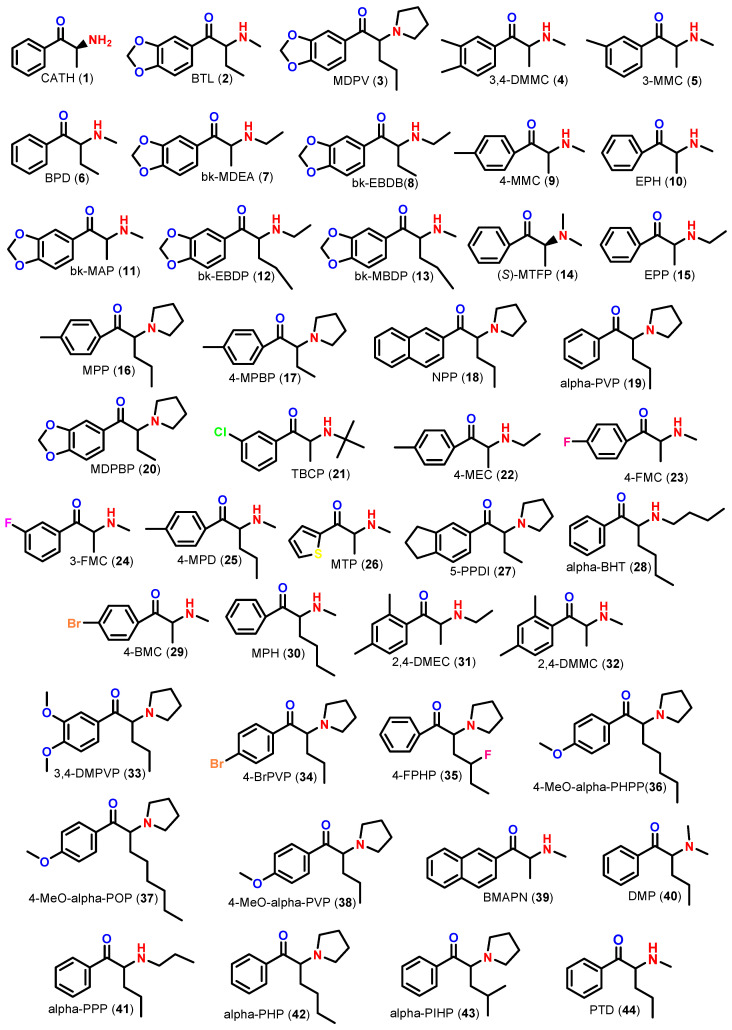
Chemical structures of 44 emergent SC selected for in silico studies, namely: CATH: cathinone or norephedrone (**1**); BTL: butylone (**2**); MDPV: 3,4-methylenedioxypyrovalerone (**3**); 3,4-DMMC: 3,4-dimethylmethcathinone (**4**); 3-MMC: 3-methylmethcathinone (**5**); BPD: buphedrone (**6**); bk-MDEA: ethylone (**7**); bk-EBDB: eutylone (**8**); 4-MMC: mephedrone or 4-methylmethcathinone (**9**); EPH: methcathinone or ephedrone (**10**); bk-MAP: methylone (**11**); bk-EBDP: *N*-ethylpentylone (**12**); bk-MBDP: pentylone (**13**); (*S*)-MTFP: (*S*)-metamfepramone or *N*,*N*-dimethylcathinone (**14**); EPP: ethcathinone (**15**); MPP: pyrovalerone (**16**); 4-MPBP: 4-methyl-α-pyrrolizinobutyrophenone (**17**); NPP: naphthylpyrovalerone or naphyrone (**18**); α-PVP: α-pyrrolidinovalerophenone (**19**); MDPBP: 3,4-methylenedioxy-α-pyrrolidinobutyrophenone (**20**); TBCP: bupropion or amfebutamone (**21**); 4-MEC: 4-methylethcathinone (**22**); 4-FMC: 4-fluoromethcathinone or flephedrone (**23**); 3-FMC: 3-fluoromethcathinone or 3-flephedrone (**24**); 4-MPD: 4-methylpentedrone (**25**); MTP: thiothinone (**26**); 5-PPDI: indanyl-α-pyrrolidinobutiophenone (**27**); α-BHP: α-butylaminohexanophenone (**28**); 4-BMC: 4-bromomethcathinone or brephedrone (**29**); MPH: hexedrone (**30**); 2,4-DMEC: 2,4-dimethylethcathinone (**31**); 2,4-DMMC: 2,4-dimethylmethcathinone or 2-methylmephedrone (**32**); 3,4-DMPVP: 3,4-dimethoxy-α-pyrrolidinopentiophenone (**33**); 4-BrPVP: 4-bromo-α-pyrrolidinopentiophenone (**34**); 4-FPHP: 4-fluoro-α-pyrrolidinohexanophenone (**35**); 4-MeO-α-PHPP: 4-methoxy-α-pyrrolidinoheptanophenone (**36**); 4-MeO-α-POP: 4-methoxy-α-pyrrolidinooctanophenone (**37**); 4-MeO-α-PVP: 4-methoxy-α-pyrrolidinovalerophenone (**38**); BMAPN: 2-(methylamino)-1-(naphthalen-2-yl)propan-1-one (**39**); DMP: dimethylpentedrone (**40**); α-PPP: α-propyloaminopentiophenone or *N*-propylpentedrone (**41**); α-PHP: α-pyrrolidinohexanophenone (**42**); α-PIHP: α-pyrrolidinoisohexanophenone (**43**); PTD: pentedrone (**44**).

**Figure 2 molecules-28-02899-f002:**
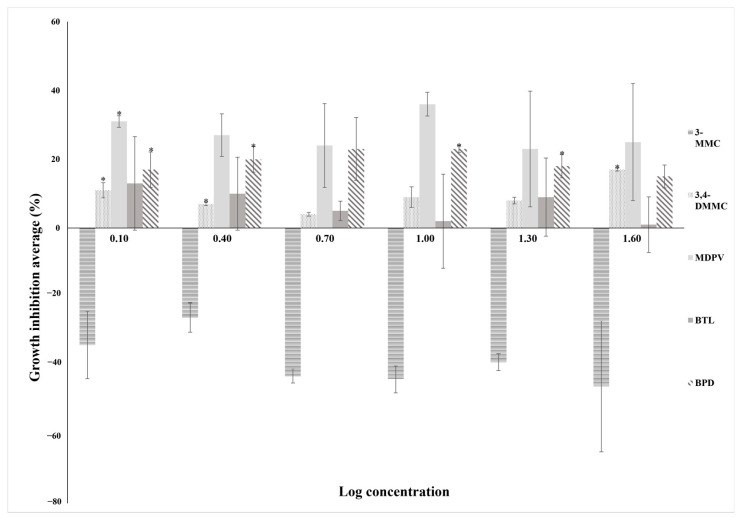
Percentage of growth inhibition vs log concentration in *Tetrahymena thermophila* after 28 h of exposure to the five SC (BPD, 3-MMC, 3,4-DMMC, MDPV and BTL) at the six concentrations tested. The results are expressed as the mean ± standard deviation (SD) obtained from three independent experiments. (Asterisks (*) represent significant differences compared to the control).

**Figure 3 molecules-28-02899-f003:**
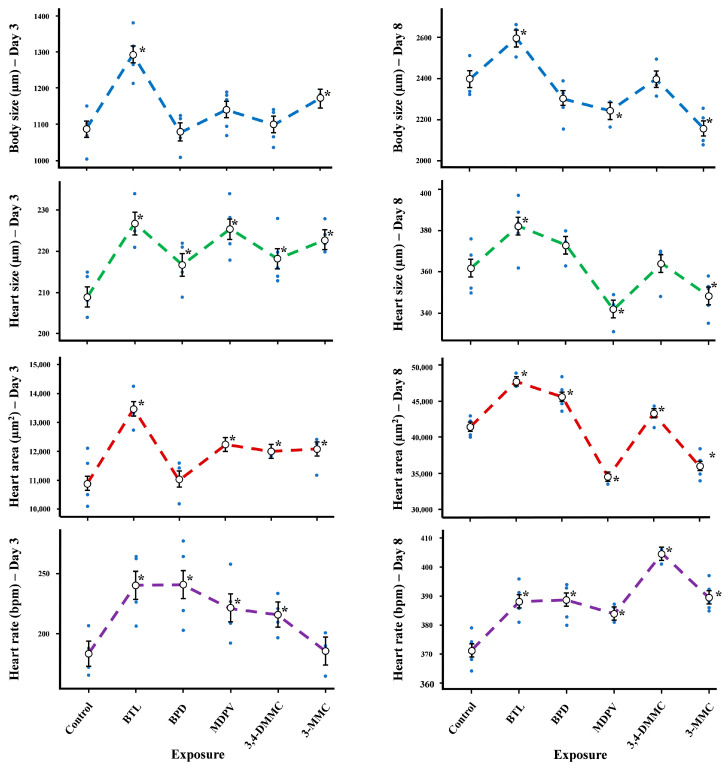
Morphophysiological effects (body size, heart size, heart area, and heart rate) on *Daphnia magna* at days 3 and 8 of exposure to BTL, 3,4-DMMC, 3-MMC, MDPV, and BPD. (Asterisks (*) represent significant differences compared to the control).

**Figure 4 molecules-28-02899-f004:**
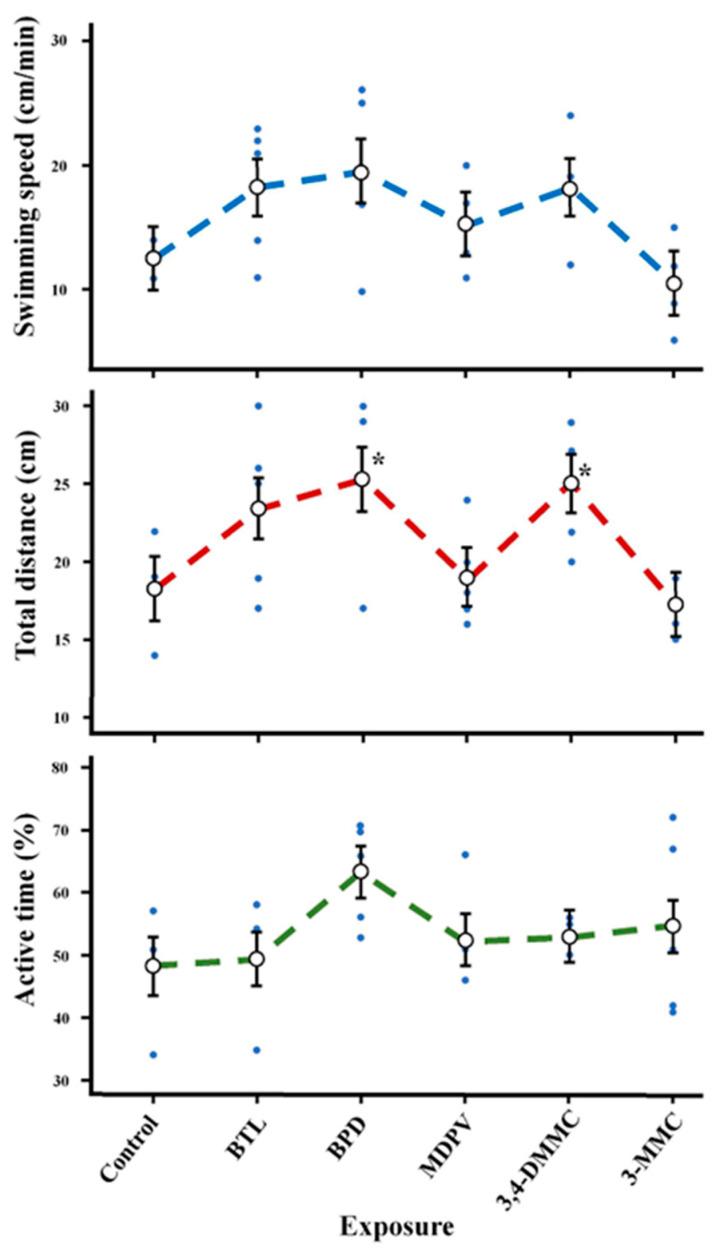
Behavioral effects (swimming speed, total distance travelled and active time) on *Daphnia magna* at day 5 of exposure to BTL, 3,4-DMMC, 3-MMC, MDPV, and BPD. (Asterisks (*) represent significant differences compared to the control).

**Figure 5 molecules-28-02899-f005:**
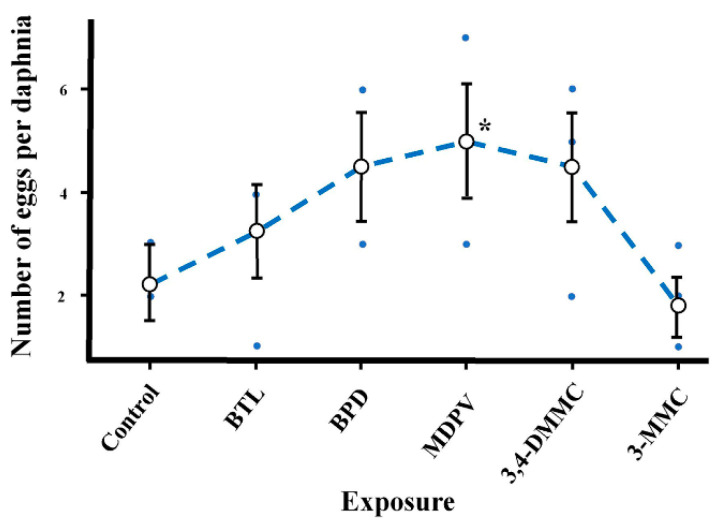
Reproductive effects (number of eggs per daphnia) on *Daphnia magna* at day 8 of exposure to BTL, 3,4-DMMC, 3-MMC, MDPV, and BPD. (Asterisks (*) represent significant differences compared to the control).

**Figure 6 molecules-28-02899-f006:**
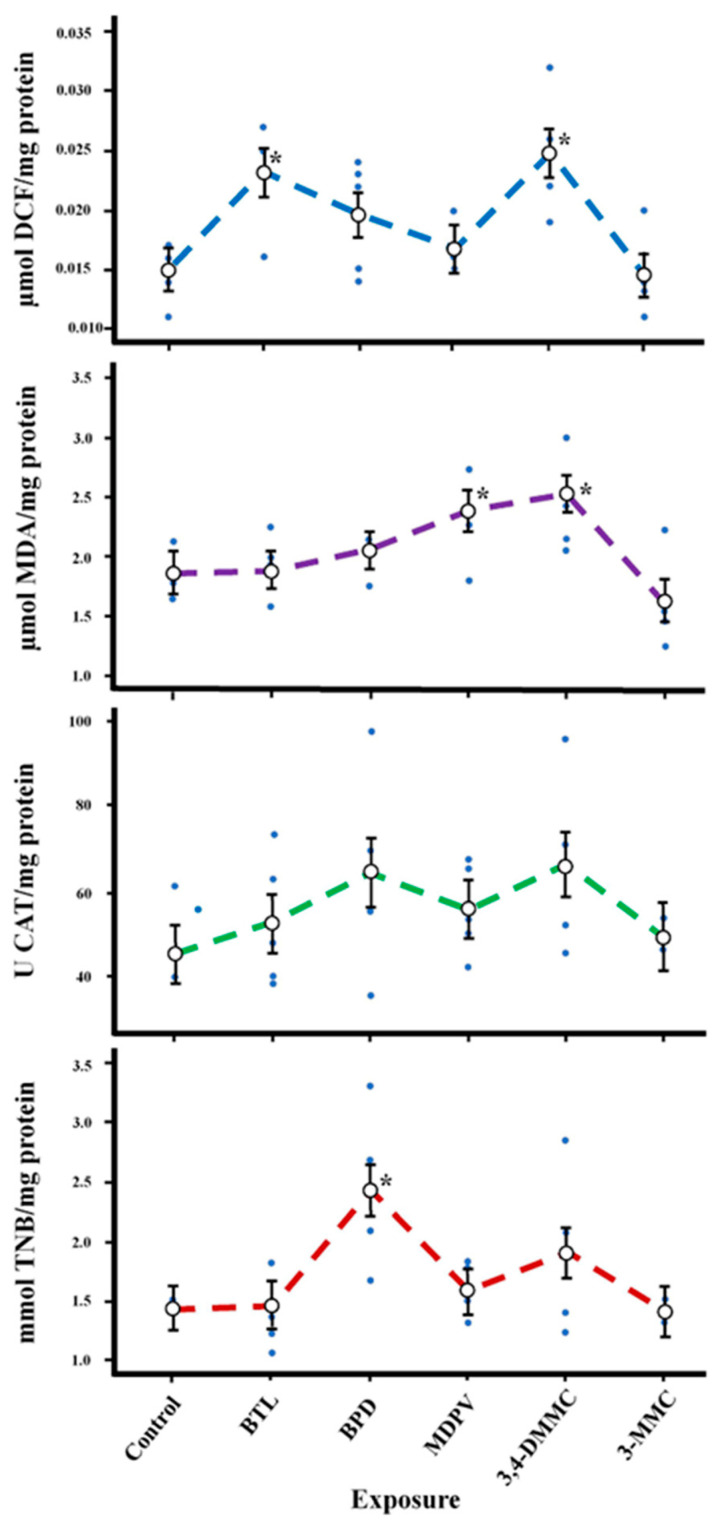
Biochemical effects (ROS, TBARS, CAT, and AChE) on *Daphnia magna* at day 8 of exposure to BTL, 3,4-DMMC, 3-MMC, MDPV, and BPD. (Asterisks (*) represent significant differences compared to the control).

**Table 1 molecules-28-02899-t001:** Predicted physical-chemical properties (i.e., WSol and log K_OW_) and toxicity data for 44 SC using EPI Suite^TM^ program (green algae, daphnia, and fish) and TEST^TM^ program (protozoan, *Tetrahymena pyriformis*).

SC	EPI Suite^TM^ Program										TEST^TM^ Program
	M_r_ (g mol^−1^)	WSKOWWIN^TM^	KOWWIN^TM^	ECOSAR^TM^			BCFBAF^TM^				
		WSol (mg L^−1^ at 25 °C)	Log K_OW_	Organism	Duration and Test	Predicted (mg L^−1^)	Estimated BCF ^1^ (L Kg^−1^ wet wt^−1^)	Estimated BAF ^1^ (L Kg^−1^ wet wt^−1^)	Estimated BCF ^2^ (L Kg^−1^ wet wt^−1^)	Estimated BAF ^2^ (L Kg^−1^ wet wt^−1^)	Predicted 48 h IGC_50_ (mg L^−1^)
**CATH**	149.19	51,470.00	1.38	Fish Daphnia Green algae	96 h/LC50 48 h/LC50 96 h/EC50	73.014 8.100 7.705	2.947 (UT) 2.407 (MT) 2.257 (LT)	2.947(UT) 2.407 (MT) 2.257 (LT)	3.469 (UT)	3.510 (UT)	148.21
				Fish Daphnia Green algae	ChV ChV ChV	5.292 0.620 2.429					
**BTL**	221.26	3076.00	2.40	Fish Daphnia Green algae	96 h/LC50 48 h/LC50 96 h/EC50	23.181 2.895 2.178	25.650 (UT) 17.490 (MT) 15.520 (LT)	25.660 (UT) 17.500 (MT) 15.530 (LT)	27.780 (UT)	28.980 (UT)	51.80
				Fish Daphnia Green algae	ChV ChV ChV	1.171 0.249 0.748					
**MDPV**	275.35	70.24	3.97	Fish Daphnia Green algae	96 h/LC50 48 h/LC50 96 h/EC50	2.675 0.401 0.210	83.320 (UT) 104.500 (MT) 110.500 (LT)	83.320 (UT) 104.600 (MT) 111.900 (LT)	973.500 (UT)	2146.000 (UT)	6.50
				Fish Daphnia Green algae	ChV ChV ChV	0.077 0.041 0.082					
**3,4-DMMC**	191.28	1515.00	2.94	Fish Daphnia Green algae	96 h/LC50 48 h/LC50 96 h/EC50	8.832 1.175 0.780	74.450 (UT) 54.140 (MT) 48.510 (LT)	74.460 (UT) 54.170 (MT) 48.630 (LT)	94.260 (UT)	105.800 (UT)	17.83
				Fish Daphnia Green algae	ChV ChV ChV	0.368 0.107 0.280					
**3-MMC**	177.25	5211.00	2.39	Fish Daphnia Green algae	96 h/LC50 48 h/LC50 96 h/EC50	18.731 2.338 1.761	25.300 (UT) 17.270 (MT) 15.320 (LT)	25.300 (UT) 17.270 (MT) 15.340 (LT)	27.430 (UT)	28.610 (UT)	39.03
				Fish Daphnia Green algae	ChV ChV ChV	0.948 0.201 0.604					
**BPD**	177.25	5819.00	2.34	Fish Daphnia Green algae	96 h/LC50 48 h/LC50 96 h/EC50	20.393 2.529 1.930	20.170 (UT) 14.540 (MT) 13.040 (LT)	20.170 (UT) 14.540 (MT) 13.050 (LT)	24.210 (UT)	25.150 (UT)	47.47
				Fish Daphnia Green algae	ChV ChV ChV	1.053 0.216 0.659					
**bk-MDEA**	221.26	3076.00	2.40	Fish Daphnia Green algae	96 h/LC50 48 h/LC50 96 h/EC50	23.181 2.895 2.178	25.650 (UT) 17.490 (MT) 15.520 (LT)	25.660 (UT) 17.500 (MT) 15.530 (LT)	27.780 (UT)	28.980 (UT)	51.56
				Fish Daphnia Green algae	ChV ChV ChV	1.171 0.249 0.748					
**bk-EBDB**	235.29	984.30	2.89	Fish Daphnia Green algae	96 h/LC50 48 h/LC50 96 h/EC50	11.727 1.551 1.042	72.840 (UT) 50.650 (MT) 44.970 (LT)	72.850 (UT) 50.690 (MT) 45.100 (LT)	84.040 (UT)	93.330 (UT)	27.48
				Fish Daphnia Green algae	ChV ChV ChV	0.498 0.141 0.373					
**4-MMC**	177.25	5211.00	2.39	Fish Daphnia Green algae	96 h/LC50 48 h/LC50 96 h/EC50	18.731 2.338 1.761	25.300 (UT) 17.270 (MT) 15.320 (LT)	25.300 (UT) 17.270 (MT) 15.340 (LT)	27.430 (UT)	28.610 (UT)	34.76
				Fish Daphnia Green algae	ChV ChV ChV	0.948 0.201 0.604					
**EPH**	163.22	17,810.00	1.85	Fish Daphnia Green algae	96 h/LC50 48 h/LC50 96 h/EC50	39.476 4.623 3.951	7.413 (UT) 5.429 (MT) 4.923 (LT)	7.413 (UT) 5.429 (MT) 4.924 (LT)	8.424 (UT)	8.587 (UT)	97.31
				Fish Daphnia Green algae	ChV ChV ChV	2.426 0.373 1.295					
**bk-MAP**	207.23	9572.00	1.91	Fish Daphnid Green algae	96 h/LC50 48 h/LC50 96 h/EC50	45.640 5.384 4.535	9.078 (UT) 6.339 (MT) 5.689 (LT)	9.078 (UT) 6.340 (MT) 5.691 (LT)	9.577 (UT)	9.777 (UT)	74.66
				Fish Daphnia Green algae	ChV ChV ChV	2.744 0.438 1.494					
**bk-EBDP**	249.31	313.90	3.38	Fish Daphnia Green algae	96 h/LC50 48 h/LC50 96 h/EC50	5.911 0.828 0.497	194.700 (UT) 144.200 (MT) 129.600 (LT)	194.800 (UT) 144.600 (MT) 130.900 (LT)	257.000 (UT)	338.900 (UT)	17.68
				Fish Daphnia Green algae	ChV ChV ChV	0.211 0.079 0.185					
**bk-MBDP**	235.29	984.30	2.89	Fish Daphnia Green algae	96 h/LC50 48 h/LC50 96 h/EC50	11.727 1.551 1.042	72.840 (UT) 50.650 (MT) 44.970 (LT)	72.850 (UT) 50.690 (MT) 45.100 (LT)	84.040 (UT)	93.330 (UT)	25.78
				Fish Daphnia Green algae	ChV ChV ChV	0.498 0.141 0.373					
**(S)-MTFP**	177.25	10,090.00	2.06	Fish Daphnia Green algae	96 h/LC50 48 h/LC50 96 h/EC50	31.145 3.738 3.043	5.591 (UT) 5.416 (MT) 5.239 (LT)	5.591 (UT) 5.416 (MT) 5.239 (LT)	13.140 (UT)	13.470 (UT)	94.97
				Fish Daphnia Green algae	ChV ChV ChV	1.776 0.309 1.015					
**EPP**	177.25	5819.00	2.34	Fish Daphnia Green algae	96 h/LC50 48 h/LC50 96 h/EC50	20.393 2.529 1.930	20.170 (UT) 14.540 (MT) 13.040 (LT)	20.170 (UT) 14.540 (MT) 13.050 (LT)	24.210 (UT)	25.150 (UT)	28.95
				Fish Daphnia Green algae	ChV ChV ChV	1.053 0.216 0.659					
**MPP**	245.37	39.83	4.46	Fish Daphnia Green algae	96 h/LC50 48 h/LC50 96 h/EC50	1.144 0.181 0.085	117.200 (UT) 154.200 (MT) 166.500 (LT)	117.200 (UT) 155.000 (MT) 174.000 (LT)	2790.000 (UT)	13,000.000 (UT)	4.33
				Fish Daphnia Green algae	ChV ChV ChV	0.028 0.020 0.035					
**4-MPBP**	231.34	124.80	3.97	Fish Daphnia Green algae	96 h/LC50 48 h/LC50 96 h/EC50	2.267 0.340 0.178	81.370 (UT) 102.100 (MT) 108.000 (LT)	81.370 (UT) 102.300 (MT) 109.400 (LT)	961.200 (UT)	2104.000 (UT)	6.55
				Fish Daphnia Green algae	ChV ChV ChV	0.066 0.035 0.070					
**NPP**	281.40	7.25	5.09	Fish Daphnia Green algae	96 h/LC50 48 h/LC50 96 h/EC50	0.507 0.086 0.035	373.800 (UT) 497.600 (MT) 540.200 (LT)	375.900 (UT) 535.500 (MT) 724.200 (LT)	9086.000 (UT)	149,800.000 (UT)	1.11
				Fish Daphnia Green algae	ChV ChV ChV	0.010 0.010 0.015					
**α-PVP**	231.34	139.40	3.91	Fish Daphnia Green algae	96 h/LC50 48 h/LC50 96 h/EC50	2.468 0.367 0.195	32.920 (UT) 43.170 (MT) 46.600 (LT)	32.920 (UT) 43.190 (MT) 46.910 (LT)	847.900 (UT)	1735.000 (UT)	6.54
				Fish Daphnia Green algae	ChV ChV ChV	0.073 0.037 0.076					
**MDPBP**	261.32	221.50	3.48	Fish Daphnia Green algae	96 h/LC50 48 h/LC50 96 h/EC50	5.336 0.756 0.443	54.280 (UT) 62.650 (MT) 64.060 (LT)	54.280 (UT) 62.670 (MT) 64.290 (LT)	321.800 (UT)	449.500 (UT)	13.42
				Fish Daphnia Green algae	ChV ChV ChV	0.184 0.073 0.167					
**TBCP**	239.75	140.20	3.85	Fish Daphnia Green algae	96 h/LC50 48 h/LC50 96 h/EC50	2.786 0.412 0.222	573.300 (UT) 424.800 (MT) 381.700 (LT)	579.600 (UT) 435.800 (MT) 399.700 (LT)	747.100 (UT)	1435.000 (UT)	4.61
				Fish Daphnia Green algae	ChV ChV ChV	0.084 0.042 0.086					
**4-MEC**	191.28	1692.00	2.89	Fish Daphnia Green algae	96 h/LC50 48 h/LC50 96 h/EC50	9.616 1.271 0.855	71.780 (UT) 49.950 (MT) 44.370 (LT)	71.790 (UT) 49.990 (MT) 44.490 (LT)	82.960 (UT)	92.020 (UT)	16.91
				Fish Daphnia Green algae	ChV ChV ChV	0.409 0.115 0.306					
**4-FMC**	181.21	9860.00	2.05	Fish Daphnia Green algae	96 h/LC50 48 h/LC50 96 h/EC50	32.366 3.880 3.166	12.320 (UT) 8.423 (MT) 7.509 (LT)	12.320 (UT) 8.424 (MT) 7.514 (LT)	12.840 (UT)	13.160 (UT)	54.94
				Fish Daphnia Green algae	ChV ChV ChV	1.853 0.320 1.055					
**3-FMC**	181.21	9860.00	2.05	Fish Daphnia Green algae	96 h/LC50 48 h/LC50 96 h/EC50	32.366 3.880 3.166	12.320 (UT) 8.423 (MT) 7.509 (LT)	12.320 (UT) 8.424 (MT) 7.514 (LT)	12.840 (UT)	13.160 (UT)	59.07
				Fish Daphnia Green algae	ChV ChV ChV	1.853 0.320 1.055					
**4-MPD**	205.30	546.60	3.38	Fish Daphnia Green algae	96 h/LC50 48 h/LC50 96 h/EC50	4.910 0.687 0.413	191.700 (UT) 142.100 (MT) 127.800 (LT)	191.700 (UT) 142.500 (MT) 129.000 (LT)	253.700 (UT)	333.500 (UT)	7.69
				Fish Daphnid Green algae	ChV ChV ChV	0.175 0.066 0.154					
**MTP**	169.25	23,770.00	1.67	Fish Daphnia Green algae	96 h/LC50 48 h/LC50 96 h/EC50	53.728 6.162 5.488	5.364 (UT) 3.959 (MT) 3.612 (LT)	5.364 (UT) 3.960 (MT) 3.612 (LT)	5.871 (UT)	5.964 (UT)	31.57
				Fish Daphnia Green algae	ChV ChV ChV	3.519 0.488 1.772					
**5-PPDI**	257.38	14.53	4.89	Fish Daphnia Green algae	96 h/LC50 48 h/LC50 96 h/EC50	0.621 0.104 0.044	42.730 (UT) 58.090 (MT) 63.790 (LT)	42.730 (UT) 58.400 (MT) 68.350 (LT)	6543.000 (UT)	71,490.000 (UT)	5.30
				Fish Daphnia Green algae	ChV ChV ChV	0.013 0.012 0.019					
**α-BHP**	247.38	20.05	4.79	Fish Daphnia Green algae	96 h/LC50 48 h/LC50 96 h/EC50	0.693 0.114 0.050	793.200 (UT) 961.100 (MT) 1001.000 (LT)	805.400 (UT) 1047.000 (MT) 1263.000 (LT)	5456.000 (UT)	48,620.000 (UT)	2.36
				Fish Daphnia Green algae	ChV ChV ChV	0.015 0.013 0.021					
**4-BMC**	242.12	1223.00	2.74	Fish Daphnia Green algae	96 h/LC50 48 h/LC50 96 h/EC50	15.235 1.979 1.378	54.330 (UT) 36.800 (MT) 32.530 (LT)	54.330 (UT) 36.830 (MT) 32.610 (LT)	59.250 (UT)	64.050 (UT)	13.81
				Fish Daphnia Green algae	ChV ChV ChV	0.683 0.177 0.487					
**MPH**	205.30	610.50	3.32	Fish Daphnia Green algae	96 h/LC50 48 h/LC50 96 h/EC50	5.346 0.743 0.452	131.600 (UT) 108.600 (MT) 100.100 (LT)	131.600 (UT) 108.700 (MT) 100.600 (LT)	223.300 (UT)	285.200 (UT)	7.42
				Fish Daphnia Green algae	ChV ChV ChV	0.195 0.071 0.168					
**2,4-DMEC**	205.30	489.40	3.43	Fish Daphnia Green algae	96 h/LC50 48 h/LC50 96 h/EC50	4.510 0.635 0.377	185.100 (UT) 147.500 (MT) 134.800 (LT)	185.200 (UT) 147.900 (MT) 136.000 (LT)	288.300 (UT)	391.000 (UT)	9.64
				Fish Daphnia Green algae	ChV ChV ChV	0.158 0.061 0.141					
**2,4-DMMC**	191.28	1515.00	2.94	Fish Daphnia Green algae	96 h/LC50 48 h/LC50 96 h/EC50	8.832 1.175 0.780	74.450 (UT) 54.140 (MT) 48.510 (LT)	74.460 (UT) 54.170 (MT) 48.630 (LT)	94.260 (UT)	105.800 (UT)	17.09
				Fish Daphnia Green algae	ChV ChV ChV	0.368 0.107 0.280					
**3,4-DMPVP**	291.39	128.90	3.56	Fish Daphnia Green algae	96 h/LC50 48 h/LC50 96 h/EC50	5.318 0.760 0.438	42.960 (UT) 52.300 (MT) 54.630 (LT)	42.960 (UT) 52.320 (MT) 54.830 (LT)	380.800 (UT)	559.200 (UT)	6.16
				Fish Daphnia Green algae	ChV ChV ChV	0.178 0.074 0.166					
**4-BrPVP**	310.24	8.65	4.80	Fish Daphnia Green algae	96 h/LC50 48 h/LC50 96 h/EC50	0.861 0.142 0.062	205.000 (UT) 272.200 (MT) 295.200 (LT)	205.200 (UT) 278.000 (MT) 333.600 (LT)	5524.000 (UT)	49,900.000 (UT)	2.23
				Fish Daphnia Green algae	ChV ChV ChV	0.019 0.016 0.026					
**4-FPHP**	263.36	23.81	4.60	Fish Daphnia Green algae	96 h/LC50 48 h/LC50 96 h/EC50	0.987 0.159 0.072	201.000 (UT) 262.000 (MT) 281.800 (LT)	201.100 (UT) 265.500 (MT) 305.000 (LT)	3752.000 (UT)	22,860.000 (UT)	1.67
				Fish Daphnia Green algae	ChV ChV ChV	0.023 0.018 0.030					
**4-MeO-α-PHPP**	289.42	8.14	4.97	Fish Daphnia Green algae	96 h/LC50 48 h/LC50 96 h/EC50	0.618 0.104 0.043	191.800 (UT) 257.900 (MT) 281.300 (LT)	192.000 (UT) 265.500 (MT) 333.900 (LT)	7541.000 (UT)	97,660.000 (UT)	2.01
				Fish Daphnia Green algae	ChV ChV ChV	0.013 0.012 0.018					
**4-MeO-α-POP**	303.45	2.57	5.47	Fish Daphnia Green algae	96 h/LC50 48 h/LC50 96 h/EC50	0.308 0.055 0.020	255.800 (UT) 348.200 (MT) 382.100 (LT)	257.400 (UT) 391.900 (MT) 640.200 (LT)	15,020.000 (UT)	56,8100.000 (UT)	1.27
				Fish Daphnia Green algae	ChV ChV ChV	0.005 0.007 0.009					
**4-MeO-α-PVP**	261.37	81.19	3.99	Fish Daphnia Green algae	96 h/LC50 48 h/LC50 96 h/EC50	2.467 0.371 0.193	88.810 (UT) 111.200 (MT) 117.500 (LT)	88.820 (UT) 111.400 (MT) 119.100 (LT)	1016.000 (UT)	2293.000 (UT)	4.85
				Fish Daphnia Green algae	ChV ChV ChV	0.071 0.038 0.076					
**BMAPN**	213.28	995.50	3.02	Fish Daphnia Green algae	96 h/LC50 48 h/LC50 96 h/EC50	8.707 1.169 0.762	104.300 (UT) 70.280 (MT) 62.010 (LT)	104.300 (UT) 70.450 (MT) 62.370 (LT)	113.400 (UT)	130.000 (UT)	5.85
				Fish Daphnia Green algae	ChV ChV ChV	0.353 0.108 0.276					
**DMP**	205.30	1059.00	3.04	Fish Daphnia Green algae	96 h/LC50 48 h/LC50 96 h/EC50	8.164 1.098 0.713	21.390 (UT) 24.400 (MT) 24.860 (LT)	21.390 (UT) 24.400 (MT) 24.880 (LT)	118.000 (UT)	135.900 (UT)	26.52
				Fish Daphnia Green algae	ChV ChV ChV	0.329 0.102 0.258					
**α-PPP**	219.33	196.30	3.81	Fish Daphnia Green algae	96 h/LC50 48 h/LC50 96 h/EC50	2.717 0.400 0.217	259.000 (UT) 249.700 (MT) 240.100 (LT)	259.100 (UT) 250.800 (MT) 244.600 (LT)	679.400 (UT)	1247.000 (UT)	5.61
				Fish Daphnia Green algae	ChV ChV ChV	0.083 0.040 0.084					
**α-PHP**	245.37	44.49	4.40	Fish Daphnia Green algae	96 h/LC50 48 h/LC50 96 h/EC50	1.245 0.196 0.093	49.750 (UT) 66.710 (MT) 72.740 (LT)	49.750 (UT) 66.840 (MT) 74.570 (LT)	2480.000 (UT)	10,450.000 (UT)	4.25
				Fish Daphnia Green algae	ChV ChV ChV	0.031 0.021 0.038					
**α-PIHP**	245.37	51.40	4.33	Fish Daphnia Green algae	96 h/LC50 48 h/LC50 96 h/EC50	1.392 0.217 0.105	44.290 (UT) 59.290 (MT) 64.610 (LT)	44.290 (UT) 59.380 (MT) 65.910 (LT)	2121.000 (UT)	7884.000 (UT)	5.29
				Fish Daphnia Green algae	ChV ChV ChV	0.036 0.023 0.042					
**PTD**	191.28	1890.00	2.83	Fish Daphnia Green algae	96 h/LC50 48 h/LC50 96 h/EC50	10.469 1.374 0.937	53.500 (UT) 40.350 (MT) 36.450 (LT)	53.500 (UT) 49.360 (MT) 36.510 (LT)	73.020 (UT)	80.130 (UT)	19.28
				Fish Daphnia Green algae	ChV ChV ChV	0.454 0.124 0.333					

BAF: bioaccumulation factor; BCF: bioconcentration factor; α-BHP: α-butylaminohexanophenone; bk-EBDB: eutylone; bk-EBDP: *N*-ethylpentylone; bk-MAP: methylone; bk-MBDP: pentylone; bk-MDEA: ethylone; BMAPN: 2-(methylamino)-1-(naphthalen-2-yl)propan-1-one; 4-BMC: 4-bromomethcathinone or brephedrone; BPD: buphedrone; 4-BrPVP: 4-bromo-α-pyrrolidinopentiophenone; BTL: butylone; CATH: cathinone or norephedrone; ChV: chronic effects values; 2,4-DMEC: 2,4-dimethylethcathinone; 2,4-DMMC: 2,4-dimethylmethcathinone or 2-methylmephedrone; 3,4-DMMC: 3,4-dimethylmethcathinone; DMP: dimethylpentedrone; 3,4-DMPVP: 3,4-dimethoxy-α-pyrrolidinopentiophenone; EC_50_: half maximal effective concentration; EPH: methcathinone or ephedrone; EPP: ethcathinone; 3-FMC: 3-fluoromethcathinone or 3-flephedrone; 4-FMC: 4-fluoromethcathinone or flephedrone; 4-FPHP: 4-fluoro-α-pyrrolidinohexanophenone; h: hours; IGC_50_: 50% of the inhibition growth concentration; LC_50_: half maximal lethal concentration; Log K_OW_: values calculated using the log octanol-water partition coefficient calculation program KOWWIN^TM^; LT: lower trophic; MDPBP: 3,4-methylenedioxy-α-pyrrolidinobutyrophenone; MDPV: 3,4-methylenedioxypyrovalerone; 4-MEC: 4-methylethcathinone; 4-MeO-α-PHPP: 4-methoxy-α-pyrrolidinoheptanophenone; 4-MeO-α-POP: 4-methoxy-α-pyrrolidinooctanophenone; 4-MeO-α-PVP: 4-methoxy-α-pyrrolidinovalerophenone; 3-MMC: 3-methylmethcathinone; 4-MMC: mephedrone or 4-methylmethcathinone; 4-MPBP: 4-methyl-α-pyrrolizinobutyrophenone; 4-MPD: 4-methylpentedrone; MPH: hexedrone; MPP: pyrovalerone; MT: mid trophic; MTP: thiothinone; M_r_: molecular weight; NPP: naphthylpyrovalerone or naphyrone; α-PHP: α-pyrrolidinohexanophenone; α-PIHP: α-pyrrolidinoisohexanophenone; 5-PPDI: indanyl-α-pyrrolidinobutiophenone; α-PPP: α-propyloaminopentiophenone or *N*-propylpentedrone; PTD: pentedrone; α-PVP: α-pyrrolidinovalerophenone; **SC:** synthetic cathinones; (*S*)-MTFP: (*S*)-metamfepramone or *N*,*N*-dimethylcathinone; TBCP: bupropion or amfebutamone; UT: upper trophic; WSol: water solubility; ^1^ Arnot-Gobas method no. 1: considering biotransformation rate estimates; ^2^ Arnot-Gobas method no. 2: assuming a biotransformation rate of zero.

**Table 2 molecules-28-02899-t002:** Percentage of growth inhibition on *Tetrahymena thermophila* after 28 h of exposure to BPD, 3-MMC, 3,4-DMMC, MDPV, and BTL.

SC Exposure/Log Concentration	Growth Inhibition Average (%)					
	0.10	0.40	0.70	1.00	1.30	1.60
BPD	17	20	23	23	18	15
3,4-DMMC	11	7	4	9	8	17
3-MMC	−34	−26	−43	−44	−39	−46
MDPV	31	27	24	36	23	25
BTL	13	10	5	2	9	1

**Table 3 molecules-28-02899-t003:** Morphophysiological effects (body size, heart size, heart area, and heart rate) on *Daphnia magna* at days 3 and 8 of exposure to BPD, BTL, 3,4-DMMC, 3-MMC, MDPV.

Variable	Day 3			Day 8		
	d.f.	*F*	*p*	d.f.	*F*	*p*
Body Size (μm)	5, 22	12.3	**<0.001**	5, 20	15.7	**<0.001**
Heart Size (μm)	5, 22	6.95	**<0.001**	5, 24	12.4	**<0.001**
Heart Area (μm^2^)	5, 23	13.9	**<0.001**	5, 24	77.8	**<0.001**
Heart Rate (bpm)	5, 20	5.08	**0.004**	5, 24	23.6	**<0.001**

d.f.: degrees of freedom; F: value of statistical test; *p*: probability (statistical differences ≤ 0.05).

**Table 4 molecules-28-02899-t004:** Behavioral effects (swimming speed, total distance traveled and active time) on *Daphnia magna* at day 5 of exposure to BTL, 3,4-DMMC, 3-MMC, MDPV, and BPD.

Variable	Day 5		
	d.f.	*F*	*p*
Swimming Speed (cm min^−1^)	5, 20	2.04	0.116
Total Distance Travelled (cm)	5, 21	3.27	**0.024**
Active Time (%)	5, 23	1.56	0.210

d.f.: degrees of freedom; F: value of statistical test; *p*: probability (statistical differences ≤ 0.05).

**Table 5 molecules-28-02899-t005:** Reproductive effects (number of eggs per daphnia) on *Daphnia magna* at day 8 of exposure to BTL, 3,4-DMMC, 3-MMC, MDPV, and BPD.

Variable	Day 8		
	d.f.	χ^2^	*p*
Number of Eggs per Daphnia	5, 19	11.5	**0.042**

d.f.: degrees of freedom; *p*: probability (statistical differences ≤ 0.05); χ^2^: value of statistical test.

**Table 6 molecules-28-02899-t006:** Biochemical effects (ROS, TBARS, CAT, and AChE) on *Daphnia magna* at day 8 of exposure to BTL, 3,4-DMMC, 3-MMC, MDPV, and BPD.

Variable	Day 8		
	d.f.	*F*	*p*
ROS (µmol DCF mg^−1^ Protein)	5, 21	4.91	**0.004**
TBARS (µmol MDA mg^−1^ Protein)	5, 21	4.32	**0.007**
CAT (U CAT mg^−1^ Protein)	5, 21	1.27	0.313
AChE (mmol TNB mg^−1^ Protein)	5, 21	3.66	**0.015**

d.f.: degrees of freedom; F: value of statistical test; *p*: probability (statistical differences ≤ 0.05).

## Data Availability

Data are available from the corresponding author upon request.

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
