# Peer review of "Integrated Approach for Synthetic Cathinone Drug Prioritization and Risk Assessment: In Silico Approach and Sub-Chronic Studies in Daphnia magna and Tetrahymena thermophila"

_molecules, 2023, doi:10.3390/molecules28072899_

Round 1

Reviewer 1 Report

The authors of the manuscript titled “Integrated approach for five synthetic cathinone drugs prioritization and risk assessment: in silico approach and sub-chronic studies in Daphnia magna and Tetrahymena thermophila”  prioritize a group of five synthetic cathinone SC for ecotoxicity studies, authors calculated log Kow of the five compounds and carried out an in vivo short-term exposures using the protozoan, Tetrahymena thermophila, and the microcrustacean, Daphnia magna, checking different indicators of toxicity across life stage.

My overall comments reject and resubmit the manuscript.

1-     The scientific value of the research is low, authors need to improve the introduction, add parts about the SC from previous literature, and increase the number of tested new psychoactive substances (NPS). The authors used only 5 compounds. The structures of SC should be represented clearly in a figure. The structures in Table 1 are very bad representations. The term in silico studies means using computer software/ simulation to study how the drugs interact with the body and with pathogens, while the manuscript uses a program to predict one term of physical properties log KOW, and toxicity. I also recommended studying the water solubility of the SC, and the total polar surface area (both parameters are important to identify the solubility of the SC compounds in the water and their absorption by living organisms).

I recommend reading this research.

https://doi.org/10.3390/molecules27072057

2-    In the results section, on page 3, line 103, the authors did not explain the results in Table 1. Authors need to give comments about table 1, after increasing the number of tested SC.

Also, much information in the results sections needs to be in experimental or discussion sections, for example, on page 3, lines 105-111. Results section to discuss the results and explain them.

Table 2 needs a control.

3-     In conclusion, the authors need to incorporate some data from the experimental part.

Author Response

The authors of the manuscript titled “Integrated approach for five synthetic cathinone drugs prioritization and risk assessment: in silico approach and sub-chronic studies in Daphnia magna and Tetrahymena thermophila” prioritize a group of five synthetic cathinone SC for ecotoxicity studies, authors calculated log KOW of the five compounds and carried out an in vivo short-term exposures using the protozoan, Tetrahymena thermophila, and the microcrustacean, Daphnia magna, checking different indicators of toxicity across life stage.

My overall comments reject and resubmit the manuscript.

Response: The authors thank the Reviewer for the careful reading of this manuscript and all the comments were considered in this revised version. We strongly believe that the comments and suggestions have increased the scientific value of the revised manuscript. We would like to take this opportunity to thank you for your contribution to maintaining the high standards of this journal.

Please see below all changes and improvements throughout the overall manuscript.

1-The scientific value of the research is low, authors need to improve the introduction, add parts about the SC from previous literature, and increase the number of tested new psychoactive substances (NPS). The authors used only 5 compounds. The structures of SC should be represented clearly in a figure. The structures in Table 1 are very bad representations. The term in silico studies means using computer software/ simulation to study how the drugs interact with the body and with pathogens, while the manuscript uses a program to predict one term of physical properties log KOW, and toxicity. I also recommended studying the water solubility of the SC, and the total polar surface area (both parameters are important to identify the solubility of the SC compounds in the water and their absorption by living organisms).

Response: We understand the reviewer concern and thank the suggestions. In this work, the term in silico means “experiment done by computer or via computer simulation”. All prediction results with the use of the appropriate software can be described as in silico studies, which means using a computer. For better clarification please see the paper: In silico toxicology protocols, Regulatory Toxicology and Pharmacology 96 (2018) 117 (https://doi.org/10.1016/j.yrtph.2018.04.014) In silico toxicology (IST) methods are computational approaches that analyse, simulate, visualize, or predict the toxicity of chemicals. IST encompasses all methodologies for analysing chemical and biological properties generally based upon a chemical structure that represents either an actual or a proposed (i.e., virtual) chemical.Today, in silico approaches are often used in combination with other toxicity tests; however, the approaches are starting to be used to generate toxicity assessment information with less need to perform any in vitro or in vivo studies depending on the decision context. IST uses models which can be encoded within software tools to predict the potential toxicity of a chemical and in some situations to quantitatively predict the toxic dose or potency. These models are based on experimental data, structure-activity relationships, and scientific knowledge (such as structural alerts reported in the literature).

To improve the quality and amount of the results of the manuscript, 39 SC were included in the in silico approaches, taking into account the EMCDDA reports and two recent publications: Bade et al., 2022 (https://doi.org/10.1021/acs.estlett.1c00807) and Almeida et al., 2022 (https://doi.org/10.3390/molecules27072057).

Please see page 3, lines 112-113: “Based on the recent reports by EMCDDA [4, 7, 22], Bade et al., (2022) [1] and Almeida et al., (2022) [29], 44 emergent SC were selected for the in silico approaches (Figure 1).”.

Still, considering these data, the INTRODUCTION was improved:

Please see page 2, lines 58-65: “Within the NPS, special attention has been given to SC, like mephedrone or 4-methylmethcathinone (4-MMC), methylone (bk-MAP), methcathinone or ephedrone (EPH), pentedrone (PTD), butylone (BTL), 3,4-dimethylmethcathinone (3,4-DMMC), buphedrone (BPD), 3,4-methylenedioxypyrovalerone (MDPV), 3-methylmethcathinone (3-MMC) [6, 7]. SC are β-keto analogs of commonly abused substances such as cathinone (CATH), isolated from Khat plant (Catha edulis), which produce similar effects to their non-keto analogs’ amphetamine type substances (indirect agonists of dopamine, serotonin, and noradrenaline receptors) [9].”.

Please see page 2, lines 87-95: “Today, in silico approaches are often used in combination with other toxicity tests to evaluate the environmental risk. Based on experimental data, structure-activity relationships, and scientific knowledge, specific software tools can be used to predict the potential toxicity and, in some situations, to quantitatively predict the toxic dose or potency. This avoids the realization of numerous in vivo assays, following European legislation, namely the Directive 2010/63/EU, firmly based on the principle of the Three Rs, to replace, reduce and refine the use of animals (vertebrates and cephalopods) used for scientific purposes. In this study, 44 SC were targeted for in silico approaches and a group of five emerging SC was prioritized for further in vivo ecotoxicity studies.”

To better clarify and illustrated the in silico data, Table 1 was greatly improved including water solubility (WSol) values, as well as, the remaining in silico results (please see pages 5-9). Also, two novel Tables were added in the SUPPLEMENTARY MATERIAL, that is Table 1S and Table 2S:

  • Table 1S included the following information: acronym, International Union of Pure and Applied Chemistry (IUPAC) name, Chemical Abstracts Service (CAS) number, chemical structure, Simplified Molecular Input Line Entry System (SMILES) notation and polar surface area (PSA) value of 44 SC for in silico
  • Table 2S included a summary of predicted data (physical-chemical properties (i.e., WSol and log KOW) and toxicity) for 44 SC using EPI SuiteTM program (green algae, daphnid and fish) and TESTTM program (protozoan, Tetrahymena pyriformis). Yet, the novel Figure 1 was introduced in the manuscript with 44 SC chemical structures (page 4).

In fact, nowadays many software’s are available to conduct in silico studies. In this work, we used the EPI SuiteTM program developed by the USEPA with KOWWINTM, WSKOWWINTM, ECOSARTM and BCFBAFTM programs. The EPI SuiteTM program uses a single input to run diverse validated estimation programs allowing the prediction of log KOW, WSol, bioaccumulation and estimate toxicity for fish (96 h and 14 days), daphnid (48 h and 21 days) and green algae (48 h), and ChV for fish, daphnid and green algae (Chlorophyta). Since no information for protozoan can be obtained using EPI SuiteTM program, the TESTTM program was executed. This application was developed by USEPA to estimate toxicity values for several endpoints including the 48 h assay on the protozoan T. pyriformis by accessing the 50 % of the growth inhibition concentration (IGC50). Considering all these new improvements, the in silico study results were therefore adjusted, please see page 3 (lines 112-118), pages 5-9 and pages 9-10 (lines 186-251). Yet, the in silico study in MATERIALS AND METHODS section was also upgraded, please see pages 19-20 (lines 488-521).

Although the number of SC has been enlarged for the in silico assays, the same is not possible with the experimental work, since it makes the experimental test impossible for comparison and humanly not feasible. Thus, and considering the aim of our work, only the five most consumed SC were selected. To clarify, please see page 3, lines 112-113: “Based on the recent reports by EMCDDA [4, 7, 22], Bade et al., (2022) [1] and Almeida et al., (2022) [29], 44 emergent SC were selected for the in silico approaches (Figure 1).” and page 9, lines 188-190: “The current study prioritized five SC for the in vivo ecotoxicity studies, namely 3,4-DMMC, BTL, 3-MMC, BPD and MDPV, based on in silico approaches data, consumption levels, EMCDDA reports [4, 22] and recent reports in wastewaters [1].”.

I recommend reading this research. https://doi.org/10.3390/molecules27072057

Response: The authors understand the comment and this article was previously included in our manuscript. The data from this paper were used to select the new 39 SC added for the in silico approaches. Please see page 3, Lines 112-113: “Based on the recent reports by EMCDDA [4, 7, 22], Bade et al., (2022) [1] and Almeida et al., (2022) [29], 44 emergent SC were selected for the in silico approaches (Figure 1).”.

2-In the results section, on page 3, line 103, the authors did not explain the results in Table 1. Authors need to give comments about table 1, after increasing the number of tested SC.

Also, much information in the results sections needs to be in experimental or discussion sections, for example, on page 3, lines 105-111. Results section to discuss the results and explain them.

Response: The authors agree with the comments. For that, these sections were adjusted and improved taking into account the introduction of 39 SC which increased the number of SC studies and improve the quality of the study (selected the five most harmful and promise SC to perform the ecotoxicity in vivo assays, considering the in silico data, the consumption levels and their presence in wastewaters). Please see all the improvement along the manuscript.

3-Table 2 needs a control.

Response: The authors thank the comment but it is a possible misunderstanding, once the data present in Table 2 concerns the in silico data obtained through TESTTM program develop by USEPA. This software allows to estimate the toxicity values for the 48 h assay on the protozoan T. pyriformis by accessing the IGC50. The Table 2 was removed from the manuscript and data were included in the Table 1 (please see pages 5-9).

4-In conclusion, the authors need to incorporate some data from the experimental part.

Response: The authors agree with the suggestion. So, the CONCLUSION was upgraded, please see page 22, lines 626-632: “Our work shows that SC does not affect mortality at the concentrations studied, however, interfered at sublethal levels with several endpoints in D. magna, precisely, morphophysiology (BTL significantly increase the body size, heart size, heart area and heart rate in juveniles and adults daphnids), swimming behavior (BPD and 3,4-DMMC significantly raise the total distance travelled), reproduction (MDPV significantly increase the number of eggs per daphnia) and oxidative stress and biochemical activity (3,4-DMMC and MDPV significantly increase the TBARS levels).”.

Reviewer 2 Report

Interesting work on the problem of water pollution with drugs and the impact of these compounds on aquatic organisms. The tested organic compounds are undoubtedly within the scope of interest of the Molecules journal, although the manuscript also has a strongly defined aspect of research related to environmental protection.

The following comments emerged:

EPI Suite program needs SMILES codes as compound input, Used SMILES of compounds should be provided in Table 1, regardless of the structure, names and CAS number, which were provided in the table..

In my opinion, it is not enough to just write in the Methods that SMILES were found on PubChem.

Entering these codes in the table will make it easier to check the researchers' calculations using the EPI Suite tools.

ECOSAR returns results for green algae, not algae, please complete this for accuracy.

Mentioning only "algae" refers to a very wide taxonomic group of organisms, green algae is a narrower group:

Archaeplastida

Viridiplantae/green algae

Mesostigmatophyceae

Chloroxybophyceae

Chlorophyta

charophyta

Rhodophyta (red algae)

Glaucophyta

Chlorarachniophytes

Euglenids

heterokont

Bacillariophyceae (Diatoms)

Axodines

Bolidomonas

Eustigmatophyceae

Phaeophyceae (brown algae)

Chrysophyceae (golden algae)

Raphidophyceae

Synurophyceae

Xanthophyceae (yellow-green algae)

Cryptophyta

dinoflagellate

Haptophyta

In the Conclusions, the authors do not refer to the results of the study of oxidative stress and biochemical activity by measuring lipid peroxidation (TBARS), ROS enzymes CAT and AChE. 

Moreover, in the Methods, I did not find a description of these methods at all. This part should definitely be improved.

Statistically, I'm not sure if 5 independent replicates are enough for this study. I also do not quite understand why d.f. there is no integer value, perhaps it has to do with the statistical tests used. In classic tests, e.g. of a t-student, d.f. is equal to n-1 if we have a paired test (testing the impact of a factor set by the researcher), so here it would be 4.

Author Response

REVIEWER 2#

Interesting work on the problem of water pollution with drugs and the impact of these compounds on aquatic organisms. The tested organic compounds are undoubtedly within the scope of interest of the Molecules journal, although the manuscript also has a strongly defined aspect of research related to environmental protection.

Response: The authors show gratitude to the Reviewer for the time to the exhaustive reading of this manuscript and all the comments. We strongly believe that the valuable comments and suggestions helped us to increase the scientific quality of the manuscript. Please see all changes and improvements throughout the overall manuscript.

The following comments emerged:

1-EPI Suite program needs SMILES codes as compound input, Used SMILES of compounds should be provided in Table 1, regardless of the structure, names and CAS number, which were provided in the table. In my opinion, it is not enough to just write in the Methods that SMILES were found on PubChem. Entering these codes in the table will make it easier to check the researchers' calculations using the EPI Suite tools.

Response: The authors thank the reviewer for the comment. For that, to better clarify the in silico data, Table 1 was changed and included the water solubility (WSol) values, as well as the remaining in silico results (Please see pages 5-9). Also, Table 1S was added in the SUPPLEMENTARY MATERIAL that included the following information: acronym, International Union of Pure and Applied Chemistry (IUPAC) name, Chemical Abstracts Service (CAS) number, chemical structure, Simplified Molecular Input Line Entry System (SMILES) notation and polar surface area (PSA) value of 44 SC for in silico studies.

2-ECOSAR returns results for green algae, not algae, please complete this for accuracy. Mentioning only "algae" refers to a very wide taxonomic group of organisms, green algae is a narrower group: Archaeplastida, Viridiplantae/green algae, Mesostigmatophyceae, Chloroxybophyceae, Chlorophyta, charophyta, Rhodophyta (red algae), Glaucophyta, Chlorarachniophytes, Euglenids, heterokont, Bacillariophyceae (Diatoms), Axodines, Bolidomonas, Eustigmatophyceae, Phaeophyceae (brown algae), Chrysophyceae (golden algae), Raphidophyceae, Synurophyceae, Xanthophyceae (yellow-green algae), Cryptophyta, dinoflagellate, Haptophyta.

Response: The authors agree with the reviewer, so the manuscript was adjusted considering the suggestion. To clarify please see page 19, lines 490-493: “The EPI SuiteTM program uses a single input to run diverse validated estimation programs allowing to predict log KOW, WSol, bioaccumulation and estimate toxicity for fish (96 h and 14 days) [31, 49, 50], daphnid (48 h and 21 days) [31] and green algae (48 h) [51], and ChV for fish, daphnid and green algae (Chlorophyta).”. Further, along the manuscript, all data referred to the term “algae” was changed to “green algae”.

3-In the Conclusions, the authors do not refer to the results of the study of oxidative stress and biochemical activity by measuring lipid peroxidation (TBARS), ROS enzymes CAT and AChE.

Response: The authors thank the reviewer for the comment and fully agree. The conclusion was improved, please see page 22, lines 626-632: “Our work shows that SC does not affect mortality at the concentrations studied, however, interfered at sublethal levels with several endpoints in D. magna, precisely, morphophysiology (BTL significantly increase the body size, heart size, heart area and heart rate in juveniles and adults daphnids), swimming behavior (BPD and 3,4-DMMC significantly raise the total distance travelled), reproduction (MDPV significantly increase the number of eggs per daphnia) and oxidative stress and biochemical activity (3,4-DMMC and MDPV significantly increase the TBARS levels).”.

4-Moreover, in the Methods, I did not find a description of these methods at all. This part should definitely be improved.

Response: The authors understand the comment, however, we think there is a misunderstanding. In fact, the description of all procedures is available in the SUPPLEMENTARY MATERIAL, in the section of Materials and methods.

5-Statistically, I'm not sure if 5 independent replicates are enough for this study. I also do not quite understand why d.f. there is no integer value, perhaps it has to do with the statistical tests used. In classic tests, e.g. of a t-student, d.f. is equal to n-1 if we have a paired test (testing the impact of a factor set by the researcher), so here it would be 4.

Response: The authors thank the reviewer for the comment. The number of independent replicates was enough, indeed, three replicates are the minimum and the experimental assays were performed with 5 replicates to increase statistical data. The d.f. abbreviations refer to statistical parameter, namely the degrees of freedom, that was defined at the end of each table. Briefly, the d.f. refers to the maximum number of logically independent values, which are values that have the freedom to vary, in the data sample. So, d.f. is calculated by subtracting one from the number of items within the data sample. For that, d.f. was five since we study five SC (each one at 10.00 µg L-1) and the control (0.00 µg L-1). The number that follows 5 refers to the residual values, which may vary depending on the values ​​considered as outliers and for that removed.

Reviewer 3 Report

The work conducted by  Pérez-Pereira al. is promising and might advance the scientific community. The authors did a good job. However, some major concerns should be addressed before this paper can be accepted for publication in Molecules. Therefore, I recommend major revisions as follows;

1-     What were the solubilities of the tested compounds? This is important in influencing the results of the ecotoxicity assays.

2-     Built on point 1, the authors should introduce a new section in the methodology detailing sample preparation and analysis.

3-     What is the chirality of the tested compounds/ Are the tested compounds have different enantiomers? If yes, this will definitely affect the toxicity of these drugs. In this case, the relative concentration of enantiomers of the chiral drugs should be detected and mentioned. 

4-     I suggest using D. magna and T. thermophila growth curves in the presence of the five synthetic cathinones to determine their differential toxicities. This involves AUCs of protozoa growth kinetics at increasing concentrations of NPS to compare overall protozoal growth versus synthetic cathinones concentration. Such estimation gives a value directly related to protozoal growth inhibition over a range of drug concentrations.

5-     To validate the findings from the toxicity assays, I suggest extending this study's results through a stable human neuronal cell line.

6-     In the results, the authors mentioned conducting a growth inhibition assay. How these assays were performed. This should be detailed in the methodology section to allow for repeating this work.

7-     Built on points 4 and 6, the following assays are missing: microplate growth assay and protozoal growth analysis to calculate the area under the curve.

8-     Please give more details in the methodology section to allow the repetition of this work.

Author Response

The work conducted by Pérez-Pereira al. is promising and might advance the scientific community. The authors did a good job. However, some major concerns should be addressed before this paper can be accepted for publication in Molecules. Therefore, I recommend major revisions as follows;

Response: The authors thank the reviewer for the careful reading of this manuscript and all the comments were considered in this revised version. We strongly believe that the comments and suggestions have increased the scientific value of the revised manuscript. Please see all changes and improvements throughout the overall manuscript.

1-What were the solubilities of the tested compounds? This is important in influencing the results of the ecotoxicity assays.

Response: The authors agree with the reviewer. To complete the information, water solubility (WSol) values were added to Table 1, including other in silico results for the 44 SC screened (Please see pages 5-9).

2-Built on point 1, the authors should introduce a new section in the methodology detailing sample preparation and analysis.

Response: The authors understand the comment, however, we think there is a misunderstanding, once the preparation of individual five SC stock solutions was present in MATERIALS AND METHODS section (4.1. Chemicals and reagents, page 19, lines 478-480). However, more information regarding sample preparation was added in the MATERIALS AND METHODS section: Section 4.3.1. Sub-chronic assay with T. thermophila, page 20, lines 529-534 (“Individual SC stock solutions were prepared at 1.00 mg mL-1 in UPW and exposure solutions by dilution of the stock solution with SFM. Concentrations used were 1.25, 2.50, 5.00, 10.00, 20.00 and 40.00 mg L-1 for each SC. For the reference test a stock solution of K2Cr2O7 at 100.00 mg L-1 in SFM was prepared and exposure concentrations by dilution with SFM. The reference test was performed at 5.60, 10.00, 18.00, 32.00 and 56.00 mg L-1.”) and Section 4.3.2.2. Experimental design, page 21, lines 555-558 (“From the individual SC stock solutions prepared at 1.00 mg mL-1 in UPW, individual intermediate solutions were prepared at 1.00 mg L-1 (in 100 mL of UPW) and stored at 4 °C, and used to prepare the concentration of 10.00 µg L-1 with MHRW for the exposure experiments.”).

3-What is the chirality of the tested compounds/ Are the tested compounds have different enantiomers? If yes, this will definitely affect the toxicity of these drugs. In this case, the relative concentration of enantiomers of the chiral drugs should be detected and mentioned.

Response: The authors thank the comment and agree. The five SC studied on in vivo ecotoxicity assays are chiral drugs and available as a racemate. In the environment (i.e., after ADMET biological process and biodegradation processes in WWTP) the SC can occur in the racemate form (50% of each enantiomer) or a different proportion of the enantiomers. Chirality is an extremely important issue and a concern to our teamwork. Indeed, from the current research, enantioselective ecotoxicity assays are being done with the most promise SC. For this study, only the racemic form used in in vivo assays correspond to 50 % of each enantiomer to prioritize further enantioselective ecotoxicity studies. To clarify this point, please see the page 19, line 474-475: “Purity of all SC standards was > 98.5 % and in the form of racemate (50.0 % of each enantiomer).”.

4-I suggest using D. magna and T. thermophila growth curves in the presence of the five synthetic cathinones to determine their differential toxicities. This involves AUCs of protozoa growth kinetics at increasing concentrations of NPS to compare overall protozoal growth versus synthetic cathinones concentration. Such estimation gives a value directly related to protozoal growth inhibition over a range of drug concentrations.

Response: Authors understand and agree fully with the reviewer. However, the construction of the dose-response curve as a function of the parameter under analysis for the microcrustacean Daphnia magna is not possible, since only a single sublethal concentration (10.00 µg L-1) was studied. In addition, daphnia estimation of mortality or immobility can be reductive as usually high concentration levels are needed to determine EC values reaching concentrations must higher than those usually found in the environment underestimating toxicity. Also, SC are psychoactive substances that affect the central nervous system. Thus, parameters such as morphophysiological, behavioral, reproductive, and biochemical parameters can be of higher relevance comparatively with traditional toxicity assays as mortality. In fact, considering in silico data, for most SC, acute toxicity data among mg L-1 levels, is considerably higher than the reported environmental concentrations. Thus, toxicity may be underestimated if only this parameter was assessed, as other parameters may be affected (as behaviour, reproduction, etc) at must lower concentrations causing a decrease in the population number of these organisms. Regarding the protozoan T. thermophila in vivo assays, the determination of EC50 or even EC20 was not possible as no relation was found between response and selected range of concentrations. Therefore, it was not feasible to determine the dose-response curves for T. thermophila, due to higher heterogeneity between concentrations.

5-To validate the findings from the toxicity assays, I suggest extending this study's results through a stable human neuronal cell line.

Response: The authors understand perfectly the comment, however at this time and taking into account our goal, we focus our research mainly on the ecotoxicity view. We agree, and it will remain a possible future endpoint to assess.

6-In the results, the authors mentioned conducting a growth inhibition assay. How these assays were performed. This should be detailed in the methodology section to allow for repeating this work.

Response: The authors understand and thank the comment, however, as we explained before, we think there is misperception, as the detailed methodology is available in the SUPPLEMENTARY MATERIAL (2.1.1).

7-Built on points 4 and 6, the following assays are missing: microplate growth assay and protozoal growth analysis to calculate the area under the curve.

Response: The authors understand and thank the comment, however, we think there is a misunderstanding, as the detailed methodology was available in the SUPPLEMENTARY MATERIAL (2.1.1).

8-Please give more details in the methodology section to allow the repetition of this work.

Response: The detailed methodology was available in the SUPPLEMENTARY MATERIAL (2.1.1).

Round 2

Reviewer 1 Report

Accept in the present form

Reviewer 2 Report

The manuscript is suitable in this form for publication in Molecules.

Reviewer 3 Report

The authors have addressed most of the raised concerns. I recommend acceptance in the current form.